# Modified Lipid Particle Recognition: A Link Between Atherosclerosis and Cancer?

**DOI:** 10.3390/biology14060675

**Published:** 2025-06-11

**Authors:** Amy E. Hall, Dhananjay Jade, Faheem Shaik, Shervanthi Homer-Vanniasinkam, Stephen P. Muench, Michael A. Harrison, Sreenivasan Ponnambalam

**Affiliations:** 1School of Molecular & Cellular Biology, University of Leeds, Leeds LS2 9JT, UK; amy23hall@gmail.com (A.E.H.); bsddj@leeds.ac.uk (D.J.); f.shaik@qmul.ac.uk (F.S.); 2School of Biomedical Sciences, University of Leeds, Leeds LS2 9JT, UK; s.p.muench@leeds.ac.uk (S.P.M.); m.a.harrison@leeds.ac.uk (M.A.H.); 3Department of Mechanical Engineering, Division of Surgery, University College London, London WC1E 6BJ, UK; s.homer-v@ucl.ac.uk; 4Department of Materials Engineering, Indian Institute of Science, Bangalore 560 012, India

**Keywords:** cancer, atherosclerosis, cancer, inflammation, signal transduction, lipid particle, scavenger receptor, LOX-1, proliferation, apoptosis

## Abstract

Cancer and atherosclerosis share molecular, cellular and pathophysiological similarities such as dysfunctional blood vessels and common risk factors including obesity, smoking and high cholesterol. Both disease states are characterized by chronic inflammation, leading to disparate effects on cell and tissue function. An emerging commonality in cancer and atherosclerosis is altered lipid particle metabolism. This raises the question: can a better understanding of lipid particle metabolism lead to new strategies to target both cancer and atherosclerosis? In this context, there are drugs that have beneficial effects on both disease states, e.g., statins. This review focuses on the molecular and cellular basis for lipid particle metabolism, highlighting common features between cancer and atherosclerosis.

## 1. Introduction

Lipoprotein or lipid particles (LPs) are unilamellar micelles composed of apolipoproteins, lipids and triglycerides which circulate in the bloodstream [1]. In *Homo sapiens*, the key LP classes (Figure 1) are categorized by their specific density [2]. Chylomicrons and very-low-density lipoprotein (VLDL) are triglyceride-rich lipid particles with low protein content, thus lighter and of low density. Low-density lipoprotein (LDL) is metabolized from VLDL and transports cholesterol to peripheral tissues to regulate membrane fluidity, signal transduction and steroid hormone biosynthesis; in contrast, high-density lipoprotein (HDL) removes excess cholesterol from tissues and returns it to the liver for excretion [3]. ApoB-100 is the main protein component of VLDL and LDL, whereas ApoB-48 and ApoA-1 are the main apolipoproteins in chylomicron particles and HDL, respectively [4].

The LDL receptor (LDLR) is key in regulating cholesterol homeostasis with LDL binding leading to endocytosis, delivery to endosomes and LDL degradation in lysosomes [5]. LDLR gene transcription is dependent on cholesterol availability; low intracellular cholesterol levels stimulate nuclear translocation of sterol regulatory element-binding protein 2 (SREBP-2), a transcription factor that promotes LDLR expression and increases LDL uptake; however, under high cholesterol levels, SRBP-2 remains cytosolic and inactive [6].

Factors such as smoking, hypertension, diabetes and high serum concentrations of LDL (e.g., familial hypercholesterolemia) increase the risk of cardiovascular disease (CVD) [7]. LDL is a key source of lipid deposits which accumulate in the innermost layer of the arterial wall, the intima, contributing to atherosclerosis, a progressive inflammatory phenomenon leading to arterial dysfunction [7]. The narrowing of arteries caused by lipid- and fat-rich plaque formation leads to hypertension and arterial disease; acute clinical end points are typified by heart attacks, strokes and limb amputations [8].

Circulating bloodstream LDL can be deposited with arterial walls during high LDL levels and/or hypertension; subsequently, arterial LDL deposits can undergo chemical modification by radical species, e.g., reactive oxygen species (ROS) [9]. Brown and Goldstein [10] first proposed that chemically modified LDL particles have pro-atherogenic properties and are a driving force in atherosclerosis. A 40-fold higher concentration of LDL (~2 mg/mL) compared to oxLDL (50 μg/mL) is needed to promote macrophage conversion into foam cells, a key hallmark of atherosclerotic plaque development [11]. Furthermore, Goldstein et al. [12] reported that LDL uptake by macrophages is too slow for foam cell generation and genetic deficiency in LDLR did not prevent cholesterol accumulation in macrophages. A new class of membrane proteins termed scavenger receptors (SRs) were proposed to mediate recognition of chemically modified or oxidized LPs. Currently, ten SR groups have been identified (Class A-J). SRs are a diverse superfamily found on immune and vascular cells, with increasing functionality in other tissues, especially under pro-inflammatory or disease states. Different SRs can discriminate between modified LPs, pathogens, phospholipids, oxLDL, acetylated LDL, malonylated LDL and apoptotic cells to mediate LP/lipid/pathogen uptake and clearance. The Class E SR is the lectin-like oxidized LDL receptor 1 (LOX-1), a 50 kDa membrane receptor which binds oxLDL and functionally linked to atherosclerosis [13,14].

Oxidation was considered the primary pro-atherogenic LP modification for decades, but recent studies indicate that multiple, highly interlinked, modifications to LP size, density and chemical composition can also occur prior to oxidation [15]. Small dense LDL (sdLDL) levels are linked to increased risk of atherosclerosis and metabolic syndrome: this LP has reduced affinity for LDLR, with increased circulatory lifetime and thus more susceptibility to chemical modification [9]. Larger sdLDL size and high affinity for intimal proteoglycans causes greater deposition within arterial walls with subsequently increased atherosclerosis [9]. Desialylation (sialic acid removal) of LDL further increases affinity for arterial proteoglycans [9]. Furthermore, elevated blood glucose levels (e.g., diabetes mellitus) promotes chemical conjugation to free ε-amino groups of lysine residues on ApoB-100 leading to glycated LDL; this increases LP susceptibility to further oxidation with increased pro-atherogenic risk [16]. Increased modified LP levels could elevate risks of both atherosclerosis and cancer [17].

LP metabolism is a critical aspect of health and homeostasis in biological organisms: such metabolism is essential for a range of catabolic, anabolic and energy-linked processes. However, LP metabolic abnormalities are linked to multiple human disease [18]. Smoking, diet and physical activity are the most common and modifiable risk factors associated with atherosclerosis and vascular dysfunction. The past 50–100 years has seen dramatic changes in human living standards, lifestyle and dietary habits linked to altered LP metabolism with impact on risk of different diseases. There is increasing evidence that the endogenous molecular clocks (circadian rhythms) coordinate temporal aspects of LP metabolism, including LP digestion, absorption in the intestine, transportation, intracellular LP metabolism and accumulation [19]. One diet type known as the Mediterranean diet is increasingly popular; this diet helps to reduce CVD morbidity and mortality. Here, a high intake of vegetables, legumes, extra virgin olive oil, nuts, fish and whole grain products, along with this moderate consumption of alcohol specifically red wine is beneficial; low intake of red meat, poultry and processed dairy products is also beneficial in this context [20,21]. Different studies show that the Mediterranean diet helps to prevent cancer, stroke and heart disease [22,23]. On the other hand, the ketogenic diet is also beneficial in disease states including CVD and cancer: here, a low carbohydrate and high fat diet is used to stimulate rapid weight loss and reduce metabolic abnormalities. The ketogenic diet promotes ketolysis, ketogenesis, and a variety of metabolic changes with favorable health outcomes in some chronic disease states [24,25]. However, there is still much debate on ketogenic diet efficacy in cancer and CVD therapy.

The Mediterranean diet is a powerful and manageable additional strategy for supporting cancer therapy by reducing intracellular oxidative and inflammatory processes, DNA damage, cell proliferation, angiogenesis, inflammation and cancer metastasis [22]. Different studies show that regular consumption of vitamin- and fiber-rich fruits and vegetables, low meat intake, and a moderate intake of milk, dairy and alcohol can enhance anti-cancer outcomes. Cellular lipid metabolism and homeostasis are regulated by sterol regulatory-element binding proteins (SREBPs) which regulate nuclear gene transcription to regulate protein expression linked to lipid metabolism. Such transcription factors are important nodes that integrate different signaling pathways and enable biological organisms to respond to a changing environment [26]. In metazoan species, the liver is a major site for processing, metabolism and synthesis of LPs, lipoproteins and lipids. Very-low-density lipoprotein (VLDL), low-density lipoprotein (LDL), intermediate density lipoprotein (IDL) and high-density lipoprotein (HDL) are different lipid or lipoprotein particles (LPs) with a single lipid monolayer and hydrophobic cores with distinct functional properties. These LPs can have common molecular components, especially lipoproteins, triglycerides, cholesterol and cholesterol esters. However, the amount and functional significance of such components are unique to a specific LP class. VLDL, IDL and LDL are pro-atherogenic factors where elevated LP levels in circulatory fluids indicate, or associate with, increased cholesterol levels and the increased risk of pathological outcomes such as heart attacks, strokes and peripheral arterial disease [27].

LDL metabolism is a well-established process in which LDL particles are recognized by the LDL receptor (LDLR), a widely expressed type I membrane glycoprotein found on many mammalian cells. Interestingly, LDL-R can bind both LDL and VLDL but enables each class of lipid particle to be internalized and trafficked through different pathways [28]. A VLDL receptor (VLDLR) which bears significant homology to LDLR, is also expressed and enables recognition of VLDL particles which exhibit ~30–80 nm diameter (Figure 1) [29]. Targeting VLDL uptake and metabolism decreases the risk of arterial disease in vivo [30]. VLDL synthesis occurs within the liver; the removal of triglycerides from VLDL by lipoprotein lipase results in both IDL (~25–35 nm diameter) and LDL (~22-28 nm diameter) (Figure 1) species, which are smaller and denser lipid particles with a higher concentration of cholesterol esters [31]. IDL has an intermediate density between VLDL and LDL. The Apolipoprotein E (*APOE*) gene encodes at least three protein isoforms, ApoE2, ApoE3 and ApoE4. Each ApoE isoform is associated with different lipid particle populations. Genetic polymorphisms within the ApoE locus are linked to hypercholesterolemia and Alzheimer’s Disease [32]. In this review, we focus on the underlying mechanisms driving modified lipid particle metabolism, signaling and implications for new disease therapies. CVD and cancer are major drivers of global mortality with high priorities for early disease diagnosis and effective therapy [33].

## 2. LOX-1 and Atherosclerosis

The transcription activator protein 1 (AP-1) binds to sites within the promoter regions of the LOX-1 (*OLR1*) and LDLR gene loci to regulate expression [34]. Endothelial cells treated with native LDL (nLDL) or oxLDL display increased phospho-ERK1/2 and phospho-p38 MAPK levels; such signal transduction also stimulates AP-1-regulated gene transcription [34]. Alongside increased LOX-1 expression, oxLDL down-regulates LDLR levels via a post-transcriptional mechanism as LDLR mRNA levels remain unchanged. One explanation for this effect is that NF-κB activation and mitochondrial DNA (mtDNA) damage are both induced by oxLDL, causing proprotein convertase subtilisin/kexin type 9 (PCSK9) secretion by endothelial and vascular smooth muscle cells; this stimulates LOX-1 expression but promotes LDLR degradation [35]. Alternatively, LOX-1 may also hinder LDLR recycling [36]; such studies require further investigation. Although NF-κB binds to the *LOX-1* (*OLR1*) promoter and upregulates gene expression, LDL does not regulate *LDLR* gene transcription similarly [34]. LDL stimulates *LDLR* gene transcription with slower kinetics compared to oxLDL-regulated LOX-1 expression; one likelihood is that SREBP2-mediated LDLR upregulation is dependent on cellular cholesterol depletion [34]. Potential interactions between AP-1 and SREBP2 in this context remain to be identified.

OxLDL binding to LOX-1 induces peroxynitrite (ONOO-) formation and NF-κB activation (Figure 2) [37]. Peroxynitrite oxidizes tetrahydrobiopterin (BH4), a cofactor for endothelial nitric oxide (NO) synthase (eNOS). The lowered BH4 levels reduces eNOS activity with a rise in superoxide levels which exacerbates oxidative stress [38]. Signaling through the PI3K-Akt pathway leads to Akt-mediated phosphorylation of eNOS-S1177 which stimulates nitric oxide production; however, Akt activation is inhibited by LOX-1-oxLDL signaling [39]. Nitric oxide prevents LDL oxidation, vascular SMC (VSMC) proliferation, platelet aggregability, monocyte adhesion and migration; reduction in nitric oxide levels promotes endothelial dysfunction and atherosclerosis [26,40]. Carbamylated LDL, which is elevated by smoking and chronic kidney disease, also activates LOX-1 leading to reduced eNOS activity and endothelial dysfunction [41]. These findings explain the increased risk of atherosclerosis caused by smoking [42].

OxLDL binding to LOX-1 activates the NADPH oxidase: association of p47phox and Rac1 GTPase on the cytosolic face of the plasma membrane bilayer promotes ROS production, including superoxide anions (O^2−^) (Figure 2) [43]. Superoxide oxidizes intracellular nitric oxide (NO) to form peroxynitrite (ONOO-), another reactive radical species. Inactive nuclear factor kappa B (NF-κB) is bound to an inhibitor and protein kinase, IκB, in the cytosol (Figure 2). Peroxynitrite triggers sustained phosphorylation, ubiquitination and degradation of IκBα; this now triggers NF-κB release, nuclear translocation and gene transcription (Figure 2) [44]. NF-κB is a transcription factor which upregulates expression of pro-inflammatory cytokines (TNF-α, IL-1, IL-6), adhesion molecules (ICAM-1, VCAM-1), chemokines (MCP-1), and pro-angiogenic signaling molecules (VEGF-A) [45]. NF-κB also binds to the 5′ regulatory region of the *OLR1* gene locus encoding LOX-1, upregulating LOX-1 expression and creating a vicious cycle of LOX-1 activation and inflammation in endothelial cells [37]. Anti-LOX-1 monoclonal antibody administration inhibits NF-κB activation and pro-inflammatory signaling [46].

Reactive oxygen species (ROS) triggers DNA damage causing endothelial progenitor cell (EPC) ageing and apoptosis; this inhibits cell proliferation, migration, adhesion and the capacity to repair the damaged endothelium lining of blood vessels [47]. The subsequent increased vascular wall permeability to LDL stimulates early events in atherosclerosis. Low-density lipoprotein (LDL) particles can undergo chemical modifications in the bloodstream (e.g., desialylation, glycation), which increases the affinity for proteoglycans in the vascular intima. Modified LDL increases caspase-3 and caspase-9 activation, causing endothelial cell apoptosis [46]. Circulating LDL particles can accumulate in the vascular intima via the damaged endothelium; such LDL deposits can undergo oxidation by ROS emanating from actively respiring and metabolically active vascular cells. LOX-1-oxLDL signaling activates the endothelium by upregulation of cell surface adhesion molecules such as intercellular adhesion molecule 1 (ICAM-1) and vascular cell adhesion molecule 1 (VCAM-1), these molecules promote monocyte adhesion to the blood vessel. The release of the chemokine monocyte chemoattractant protein-1 (MCP-1) further promotes monocyte migration into the vascular intima and subsequent differentiation into macrophages [47]. Macrophages release pro-inflammatory cytokines and ROS which further promote LDL oxidation. OxLDL stimulates LOX-1 upregulation in macrophages, causing oxLDL uptake and foam cell formation. Subsequent apoptosis of foam cells seeds formation of the necrotic core of the developing atherosclerotic plaque. Low oxLDL levels can promote vascular smooth muscle cell (VSMC) proliferation [48] with increased extracellular matrix synthesis leading to a fibrous cap over the atherosclerotic plaque [49]. However, high oxLDL levels trigger VSMC apoptosis leading to weakening of the fibrous cap during advanced stages of atherosclerosis [50]. Angiogenesis also destabilizes the plaque with increased neovessel leakage and erythrocyte-derived cholesterol accumulation [51]. OxLDL triggers the formation of neutrophil extracellular traps (NETs), a process termed NETosis. Histone H4 release in NETs induces VSMC lysis, increasing plaque instability [52]. Damage to the fibrous cap reveals pro-thrombotic factors such as tissue factor and Von Willebrand Factor (VWF) which promote blood clotting and arterial blockage [53].

The role of LOX-1 in initial stages of atherosclerosis is well-established. LOX-1 upregulation is detected in non-lesion areas of coronary arteries in the Watanabe strain of hyperlipidemic rabbit model, suggesting early role(s) in atherosclerosis [48]. Liu et al. [49] found that *miR-let-7g*, which inhibits LOX-1 expression by binding to the *LOX-1* mRNA 3′ untranslated region (UTR), suppresses VSMC proliferation and lesion size. However, this study did not consider whether *miR-let-7g* inhibits the wider expression of other pro-atherogenic factors which contribute to this miRNA’s anti-atherosclerotic properties.

### 2.1. LOX-1 RNA Splicing and Disease

Alternative splicing of the OLR1 primary RNA transcript can produce three different LOX-1 splice isoforms. The LOXIN splice isoform lacks exon 5 that encodes two-thirds of the CTLD: this prevents oxLDL recognition and downstream signaling [45]. LOXIN also forms heterodimers with full-length LOX-1, further down-regulating oxLDL binding and signaling compared to a functional LOX-1 homodimer [54]. Adenovirus-mediated LOXIN expression suppresses oxLDL-mediated apoptosis in endothelial progenitor cells; targeting OLR1 RNA splicing could thus modulate LOX-1 function linked to disease outcomes. However, much of the current focus is on direct LOX-1 inhibition as a clinical strategy [45,55].

### 2.2. LOX-1, Cancer and Lipid Metabolism

One most common biochemical characteristics of cancer cells is aberrant glucose metabolism. For mammalian cells, the main carbon and energy source is glucose. Cellular glucose uptake by glucose transporters leads to glycolytic metabolism of glucose to pyruvate, which yields a net production of two ATP molecules per glucose. In cancer cells, an increased glycolytic rate is utilized to produce ATP more rapidly (although less efficient) compared to oxidative phosphorylation in the mitochondria. This phenomenon was first described by Otto Warburg in the 1930s and is referred to as the Warburg effect or aerobic glycolysis [56]. Conversion of glucose to lactate under aerobic conditions is one of the leading hallmarks of cancer cells; this biochemical pathway is a major focus for cancer therapy.

An emerging role for LOX-1 involves oxLDL interactions promoting cancer development: LOX-1-dependent activation of canonical MAPK and NF-κB pathways could cause changes in metabolism, cell proliferation, migration and invasion [57,58]. One feature of LOX-1-oxLDL complex formation is increased ROS levels [59]. In endothelial cells which line all blood vessels, LOX-1-oxLDL signaling reduces nitric oxide levels alongside activation of NF-κB, P13K-AKT and GSK3β signaling pathways [60,61]. Gene expression events linked to epithelial-to-mesenchymal transition (EMT) and NF-κB pathways are triggered, with a rise in pro-inflammatory interleukin signaling (IL-6, IL-8, IL-1β). Although a definitive role for LOX-1 in cancer is yet to emerge, cancer-associated signatures indicate upregulation of gene expression linked to inflammation and carcinogenesis, with relevance to lipid metabolism during cellular transformation. LOX-1 upregulation in epithelial cancers [62,63] may be one avenue of anti-cancer drug resistance [64].

### 2.3. LOX-1-Specific Therapeutics

Soluble LOX-1 (sLOX-1) is generated by alternate RNA splicing, proteolytic cleavage or ‘shedding’ of the LOX-1 extracellular domain: circulating sLOX-1 could be a sensitive biomarker of CVD, MI or CAD than cardiac troponins, traditional biomarkers of myocardial infarction [65,66]. However, use of sLOX-1 to predict vascular disease including T2D status has employed non-standard and technically complex ELISAs, hindering clinical translation [67]. Elevated LOX-1 expression is linked to poor prognosis in some epithelial cancers such as prostate [68], colorectal [46] and squamous non-small cell lung cancer [69]. LOX-1 is thus a desirable therapeutic target for simultaneous targeting of atherosclerosis and cancer [70]. Natural compounds which inhibit LOX-1 expression, including gingko biloba, curcumin and ellagic acid [40] provide low-risk routes for supporting anti-cancer therapy. Chinese tea-derived polyphenols inhibit NF-κB and MAPK signaling and foam cell formation [71], indicating that dietary agents could be highly effective in this context. PLAzPC, a modified phospholipid which binds the hydrophobic tunnel within the LOX-1 extracellular domain, is one of the most effective inhibitors of oxLDL binding; this is also a highly effective anti-inflammatory molecule (Table 1) [72]. Using PLAzPC to delay oxLDL binding and uptake by LOX-1 could provide a valuable window for oxLDL clearance by host systems. The small molecule BI-0115 is a LOX-1 inhibitor that binds and stabilizes two LOX-1 dimers, causing steric inhibition of oxLDL binding and LOX-1 downstream signaling [73]. Future studies employing structure-guided screening of large chemical databases could enable identification and development of new LOX-1-specific inhibitors for dual use in atherosclerosis and cancer (Table 1).

LOX-1 is a potential therapeutic target (Figure 3) in atherosclerosis and related vascular disease states, with the LOX-1-oxLDL signaling pathways attracting detailed studies at the molecular, cellular and translational levels. LOX-1-specific humanized mAb development is problematical due to the conserved nature of the oxLDL-binding domain in mammals [40]. However, an anti-human LOX-1 chicken monoclonal antibody blocks LOX-1 activation and oxLDL uptake [74]. Preclinical studies on LOX-1 inhibitors have sparked much interest in the translation of such work to clinical use for different disease states (https://clinicaltrials.gov/ (accessed on 1 May 2025)). Clinical trials are underway using an anti-LOX-1 neutralizing mAb MED16570^®^ (AstraZeneca, Cambridge, UK). A phase I clinical trial (#NCT03654313) is assessing the safety of increasing doses of MEDI6570^®^ in subjects with type 2 diabetes mellitus whilst a phase IIB clinical trial (EudraCT #2020-000840-75) is testing whether MEDI6570^®^ reduces non-calcified atherosclerotic plaque volume or improves biomarkers of heart failure [65]. The GOLDILOX phase IIB clinical trial (#NCT04610892) involved MI patients treated with increasing doses of MEDI6570^®^; this work is completed, but published studies have not yet arisen. One issue is the frequency of anti-LOX-1 monoclonal antibody injections could cause patient non-compliance with reduced treatment efficacy [47]. Such clinical trials help to lay foundations for LOX-1 inhibitor use in vascular disease therapy. Ishino et al. have used technetium-99-labeled anti-LOX-1 antibody and molecular imaging to detect atherosclerotic lesions in arterial beds [75]. Suppression of LOX-1 expression in human coronary artery endothelial cells using antisense oligonucleotides to the 5′-coding sequence of human *OLR1*, not only blocked oxLDL-induced LOX-1 expression but also MAPK activation [76]. MicroRNA (miRNA) are also feasible approaches for lowering LOX1 expression in pathophysiological states: administration of this type of non-coding RNA of 22-26 nucleotides can modulate protein-encoding mRNA levels. The *OLR1* 3′ UTR mRNA contains a binding site for *miR-let-7g*; such miRNA administration inhibits oxLDL-induced LOX-1 expression [77].

The *miR-let-7g* not only inhibits LOX-1 expression but also oxLDL-induced VSMC migration and proliferation [77], thus supporting the idea of targeting LOX-1 function in disease states [40]. Other miRNAs have also been postulated including *miRlet-7a*, *miR-let-7b*, *miR-24*, *miR-98*, *miR-186-5p*, *miR-210*, *miR-320a*, *miR-369-3p*, *miR-590-5p* and circ*TTLL13*. More work is needed on comparing small molecules, monoclonal antibodies and nucleic acids in targeting LOX-1 function in cancer and CVD but also neurodegenerative disease. Therapies aimed at specific events associated with LOX-1-oxLDL signaling are currently under development or already licensed. These include approaches to reduce circulating LDL (e.g., statins, anti-PCSK9), prevent oxidation (antioxidants) and inhibit LOX-1 (PLAzPC, MEDI6570, *miR-let-7g*, antisense *OLR1*) (Figure 3). Pathways downstream of LOX-1 activation may also be targeted. Reactive oxygen species (ROS) production and therefore oxidative DNA damage and NF-κB activation may be inhibited by myeloperoxidase inhibitors and NADPH oxidase inhibitors. Protein-arginine deiminase type-4 (PAD4) and neutrophil elastase inhibitors prevent NET formation by aiming to reduce EMT and LDL oxidation. Inhibition of EMT (e.g., resveratrol) and angiogenesis (e.g., triptolide) (Figure 3) may prevent tumor metastasis in this context. Cancer therapies aimed at the Warburg effect (see later) have gained much attention; however, the lack of clinical success has led to search for alternatives [78]. LOX-1-specific targeting as a pro-inflammatory mediator raises the possibility of dual anti-cancer and anti-atherosclerosis therapy.

**Table 1 biology-14-00675-t001:** LOX-1 inhibitors, natural products and clinical usage.

Drug	Model	LOX-1 Modulatory Effects	Clinical Use
Statins [79,80,81]	COS cells HCAECs	Competitive binding to hydrophobic tunnel of LOX-1 ligand-binding domain	YES
Losartan [82]	SD rats	Inhibition of inflammation and apoptosis	YES
Cigitazone [83]	RMVEC	Activation of PPARγ, eNOS and AMPK	YES
Probucol [84]	HK-2 cells	Inhibition of p38 MAPK, ERK1/2, and ROS signaling	YES
Rapamycin [85]	HUVECs	Inhibition of mTOR and NF-κB signaling	YES
Curcumin [86]	HUVECs HCAECs	Inactivation of AP1 and NF-κB signaling	NO
Dihydrotanshinone I [87]	HUVECs	Inactivation of NOX4 and NF-κB signaling	NO
PLAzPC [72]	COS cells	Competitive binding to hydrophobic tunnel on LOX-1 lignd-binding domain	NO
Ursolic acid [88]	HUVECs	Inactivation of TLR4 and MyD88 signaling	NO
Hyperoside [89]	VSMCs	Inactivation of ERK1/2 signaling	NO
Quercetin [90]	RAW264.7	Inhibition of ROS generation and STAT3 signaling	NO
Resveratrol [91]	RAW264.7	Inactivation of NOX4 and ROS signaling	NO
Ginsenoside F1 [92]	HUVECs	Inactivation of NF-κB and decrease in pro-inflammatory signals	NO
Ginkgolide B [93]	HUVECs	Inactivation of NOX4 and ROS signaling	NO
Cryptotanshinone [94]	HUVECs	Inactivation of NOX4 and ROS signaling	NO
Tanshinone IIA [95]	RAW264.7	Inactivation of NF-κB signaling	NO
Berberine [96]	HUVECs	Inactivation of PI3K-AKT, ERK1/2, and p38 MAPK signaling	NO

## 3. Warburg Effect on Cancer and CVD

The Warburg effect promotes cancer development, progression and immunosuppression through multiple mechanisms [78]. Some studies show immune cell responses to inflammatory LPs such as oxLDL depend on cellular metabolic status: oxLDL is not only an oxidative stress marker, but also actively promotes inflammatory processes and interacts with various immune cells [97]. Such oxLDL involvement extends to autoimmune diseases, e.g., anti-phospholipid syndrome (APS), where β2-glycoprotein I (β2GPI) is recognized by anti-phospholipid antibodies (aPL) [98,99]. One emerging area of study is the intersection between cellular metabolism and immune function to provide a better understanding of how metabolic changes modulate chronic inflammation and disease states, especially in CVD [100,101]. There is increasing evidence of links between altered lipid metabolism to cancer risk [69,102,103,104]. Increased oxLDL levels cause cellular dysfunction and promote metabolic syndrome-related pathologies such as atherosclerosis [105,106] and non-alcoholic steatohepatitis (NASH) [107]. Changes in the intracellular environment can influence membrane ATP-regulated pump activity, with more ATP usage during disease states; here, the rapid increase in aerobic glycolysis can deliver more ATP, whereas oxidative phosphorylation-regulated ATP synthesis remains unchanged [108]. OxLDL impacts on immune cell metabolism by stimulating glycolytic ATP synthesis but has negligible effect on oxidative phosphorylation [109,110].

OxLDL binding to immune cells such as dendritic cells and macrophages, promotes signaling, increased glucose transporter and glycolytic enzyme expression, thus facilitating increased glucose uptake and glycolysis [111]. Increased pyruvate kinase, hexokinase and phosphofructokinase activity can stimulate glycolysis and ATP synthesis [112,113,114]. OxLDL stimulates pyruvate and hexokinase gene expression, thus increasing catalytic events which stimulate glycolytic flux [114]. OxLDL enhances phosphofructokinase activity in F-6-P conversion into F-1,6-P2, a rate-limiting step which has substantial effects on glycolysis [112,114,115]. OxLDL-stimulated glycolytic flux promotes rapid ATP synthesis; furthermore, oxLDL provides metabolic intermediates for anabolic pathways, e.g., amino acid and nucleotide synthesis, essential for cell function, proliferation and homeostasis. In dendritic cells, oxLDL-stimulated glycolysis supports immune cell maturation, facilitating antigen presentation to T-cells, promoting pro-inflammatory T-cell differentiation and immune response amplification [111,113]. In macrophages, oxLDL-stimulated glycolysis is linked to the increased production of pro-inflammatory cytokines leading to chronic inflammation [110,113].

Oxidative phosphorylation in mitochondria is highly regulated both at kinetic and thermodynamic levels [116]. Two non-mutually exclusive mechanisms regulate mitochondrial respiration: regulation of either enzyme kinetic rates during oxidative phosphorylation [117] or mitochondrial respiratory gene expression [118,119]. Immune cell oxLDL exposure induces significant mitochondrial dysfunction, impacting on energy production with increased ROS levels. Mitochondrial dysfunction induced by oxLDL has several implications for immune cell function. Uptake of oxLDL disrupts mitochondrial function with effects on electron transport and ATP synthesis. These include altered electron transport chain (ETC) function, collapse of the mitochondrial membrane potential (MMP), increased ROS levels, mitochondrial DNA damage. OxLDL disrupts function of ETC complexes I and III, leading to electron leakage and reduced ATP synthesis [120,121,122]. OxLDL also affects immune cell fatty acid (FA) metabolism, with changes in lipid accumulation and lipid signaling pathways. Immune SRs facilitate oxLDL-associated lipid uptake and processing leading to accumulation of oxidized lipids, particularly cholesterol ester and triglycerides; such accumulation drives lipid droplet formation and differentiation into foam cells [123,124]. Peroxisome proliferator-activated receptor (PPAR) activation by oxidized lipid and FA products modulates gene expression to influence lipid metabolism and inflammation [125,126].

Dysregulated lipid metabolism in cancer can alter membrane composition, gene expression, signaling pathways that regulate cell function and disease progression [127]. Uncontrolled cancer cell proliferation facilitates cell survival in unfavorable environments lacking oxygen and nutrients; tumor cells undergo metabolic changes which facilitate lipid accumulation and lipid oxidation [128]. Cholesterol, phospholipids and FAs are potent signaling regulators that impact ATP synthesis. Cancer cell metabolism can be rewired by lipids in the production of membranes, second messengers and ATP central to anabolic pathways necessary for cell proliferation [129,130]. Lipid metabolism during cancer progression is linked to tumor microenvironment remodeling. During cancer metastasis, lipid metabolism changes may facilitate tumor cell survival during cell migration and secondary tumor colonization; lipid-modulating agents are emerging as promising anti-cancer therapeutics [131]. The development of cancer metabolomics is providing new non-invasive diagnosis and screening tools [132,133]. Increased lipid metabolism in myeloid-derived suppressor cells (MDSCs), regulatory T-cells (Tregs), and tumor-associated macrophages (TAMs), occurs by upregulating lipid uptake and FA oxidization (FAO), facilitating immune suppressive function. Excessive lipid accumulation caused by elevated CD36 levels in CD38+ T-cells impairs secretion of anti-tumor factors (IFN-γ, TNF-α) [134,135]. CD36 upregulation on natural killer (NK) cells also impairs tumor-killing caused by intracellular lipid accumulation and cell dysfunction. Blocking CD36-mediated lipid uptake on cytotoxic CD38+ T-cells or Tregs enhances anti-tumor immune response(s) [135,136].

## 4. CD36 Function in Health and Disease

A Class B SR termed CD36 binds oxLDL; however, whether CD36 is a pro- or anti-atherogenic factor is unclear. Cellular oxLDL uptake mediated by CD36 contributes to foam cell development and promotes formation of a CD36-TLR4-TLR6 complex which promotes pro-inflammatory NF-κB signaling in macrophages [50]. However, an atheroprotective role for CD36 is also evident by removing modified phospholipids and/or lipid particles from the vascular bed [45]. Furthermore, some monocytes express CD36, which binds oxLDL, and activate Src family kinases with oncoprotein functionality, leading to altered actin dynamics and cell migration. Increased monocyte patrolling could reduce more static endothelial-monocyte interactions, thus decreasing inflammation and early pro-atherogenic events within arterial walls. The mechanism and identity of Src oncoprotein activation by CD36-oxLDL signaling pathway remains to be elucidated [51].

### 4.1. CD36 and Cancer

The Class B SR member, CD36, is heavily implicated in atherosclerosis [137], and potentially in cancer progression [138]. Amplification of the *CD36* locus is detected in many cancer metastases [139]; mouse cancer models transplanted with *CD36* KO breast cancer or melanoma cells showed reduced lung, bone and liver cancer tumor metastases [140]. It is well-established that a lipid-rich diet can promote cancer metastasis; CD36 could thus contribute to cancer progression by stimulating lipid uptake and FA β-oxidation in cellular energy metabolism [141]. Investigation of mouse scavenger receptor gene expression in metastatic tumors found that although LOX-1 is significantly upregulated, CD36 is downregulated [142]. The role of CD36 in cancer development, progression and metastasis thus requires further study. Notably, there is increased expression of EMT-associated genes in CD36-negative cells in primary tumors and lymph node metastases. One possibility is that cooperation between CD36-positive and CD36-negative cells facilitates tumor invasion for secondary tumor metastases [140].

### 4.2. CD36 Disease Therapy

CD36 is a pro-atherogenic factor in CVD; inhibition of CD36-mediated FA uptake suppresses EMT in HCC cells [143]. However, cardiomyocyte-specific *CD36* KO mice display accelerated cardiac hypertrophy leading to heart failure [53]. One problem is that CD36 inhibition could disrupt essential functionality, e.g., myocardial FA uptake needed for ATP synthesis [62]. Whether CD36-binding properties can be selectively blocked for oxLDL binding but retain FA uptake is unclear [104]. Targeting CD36 in macrophages and cancer cells could modulate disease development and progression. Ezetimbe, a clinically approved inhibitor of intestinal cholesterol uptake, down-regulates CD36 expression and foam cell formation, suppressing atherosclerosis development and progression [144]. Furthermore, tamoxifen (a breast cancer drug) blocks PPARγ nuclear translocation and inhibits CD36 gene transcription, reducing foam cell formation [145].

CD36 post-translational modification (PTM) and turnover also affects disease outcomes. CD36 undergoes ubiquitination and covalent attachment of both K48 and K63 polyubiquitin chains [146]; CD36-ubiquitin cleavage by deubiquitinases such as UCHL1 [147] or USP11 [148] modulates foam cell formation and atherosclerosis-linked outcomes. Targeting UCHL1 promotes CD36 degradation and suppresses foam cell formation [147]. UCHL1 is also overexpressed in lung adenocarcinomas, gastric cancer and myelomas [149] and deubiquitinates HIF-1α; the HIF-1αβ heterodimer binds to HREs within genes such as VEGFA to stimulate gene transcription during hypoxia [150]. Current UCHL1 inhibitors (e.g., LDN57444) lack sufficient UCHL1 selectivity to be used in disease-specific therapy [151]. Targeting CD36, UCHL1 or USP11 using small molecule inhibitors or reverse genetic strategies could provide new strategies to target both atherosclerosis and cancer.

### 4.3. CD36 Signaling in Atherosclerosis

CD36 binds oxLDL but whether CD36 is a pro- or anti-atherogenic factor is unclear. Cellular oxLDL uptake mediated by CD36 contributes to foam cell development and promotes formation of a CD36-TLR4-TLR6 complex which promotes pro-inflammatory NF-κB signaling in macrophages [50]. However, an atheroprotective role for CD36 is also evident by removing modified phospholipids and/or lipid particles from the vascular bed [45]. Furthermore, some monocytes express CD36 which binds oxLDL and activates Src family kinases with oncoprotein functionality, leading to altered actin dynamics and cell migration. Increased monocyte patrolling could reduce more static endothelial-monocyte interactions, thus decreasing inflammation and early pro-atherogenic events within arterial walls. The mechanism and identity of Src oncoprotein activation by CD36-oxLDL signaling pathway remains to be elucidated [51].

## 5. Links Between Cancer and Atherosclerosis

A meta-analysis revealed ≥2-fold cancer risk in patients with atherosclerotic CVD compared to a non-atherosclerotic CVD group over a ~3 yr period [52]. However, more information on cancer stage progression is lacking. Future studies are required which enable stratified analyses to establish the rate of cancer onset and progression in CVD cases of variable severity [52]. A retrospective study by Li et al. [53] identified coronary heart disease as an independent risk factor for cancer. However, this study relied on a relatively small human clinical sample size: 600 patients enrolled between Jan. 2012–June 2019 were divided into groups according to CAD or cancer incidence [53]; further cohort studies are thus required to better support the link between these disease states.

Notably, transcriptional profiling of isogenic models of cellular transformation by Hirsch et al. [62] detected upregulated genes including *LOX-1*, thus associating cancer with metabolic and pro-inflammatory changes especially relevant to atherosclerosis. The finding that higher plasma oxLDL levels correlate with increased risk of breast, pancreas, colon and esophageal cancer [103] indicates a mechanistic link between atherosclerosis and cancer. Chemical modification or oxidation of LDL is linked to obesity, smoking and diabetes. These clinical conditions are additional risk factors in atherosclerosis and cancer, further supporting a functional role for oxLDL linking both disease states [9].

Cellular transformation involves upregulation of pro-inflammatory (LOX-1, IL-1β, IL-6, IL-8) and hypoxia-linked (HIF-1α, VEGF-A) gene products [57]. However, the MCF-10A primary fibroblast line shows that knockdown of LOX-1 suppresses NF-κB activation and cellular transformation into an immortalized state resembling the cancer state [62]. LOX-1-regulated NF-κB activation could thus be a link between atherogenesis and cellular transformation in cancer.

In addition to pro-atherogenic outcomes, ROS can cause DNA damage which promotes either the activation of proto-oncogenes or inactivates tumor suppressor genes [144]. OxLDL treatment and ROS production upregulates NADPH oxidase levels in human mammary epithelial cells upon oxLDL administration [145]. Such findings further support role(s) for oxLDL in transformation of cells into a malignant or immortal state. In this context, administration of 4-hydroxynonenal (an oxLDL component) to rat hepatocytes increased micronuclei, chromosomal abnormalities and sister-chromatid exchanges indicating increased DNA damage [152]. Furthermore, Murdocca et al. [153] showed that LOX-1 knockdown in a colon cancer cell stimulates production of a volatile butyrate compound with epigenetic regulatory properties resulting in decreased cellular neoplasia. LOX-1 is thus implicated as a tumor-promoting factor which modulates tumor suppressor gene expression; however further studies to elucidate this mechanism are required [153]. A microarray analysis on *OLR1* KO mice shows that abrogation of *LOX-1* in tandem with inhibition of NF-κB target gene expression causes a profound inhibition of key enzymes in lipogenesis [145]. Murdocca et al. studies reveal LOX-1 upregulation during different stages of colon tumorigenesis [153], further supporting a role of LOX-1 in colon cancer. LOX-1 knockdown using RNAi in DLD1 colon cancer cells influenced butyrate levels, with a marked increase in histone H4 acetylation, suggesting a link between LOX-1 signaling and the epigenetic control of tumor suppressor gene expression [153]. The modifiable nature of epigenetic marks has provided new targets in a range of disease states. The role of epigenetics in modulating atherosclerosis is increasingly evident [154]. Such epigenetic inflammation and coronary artery disease (CAD) can be potentially reversed by inhibitors of DNA methyltransferases, histone acetyltransferases, histone deacetylases, histone methylases, and bromodomain and extra-terminal motif (BET) containing proteins. RVX-208 is a more selective BET inhibitor and is currently under investigation in phase II clinical trials for CVD [155,156]. There is thus an emerging avenue for small molecule-based targeting of chromatin architecture to combat atherosclerosis [157].

### 5.1. Cancer Progression

Diakowska et al. [103] found significantly higher oxLDL levels in early vs. advanced stage colorectal cancer (CRC) patients. One hypothesis is that oxLDL promotes early or initiating steps but not in progressive stages of cancer. Alternatively, the increased metabolic flux in rapidly dividing cancer cells could mean increased LDL uptake in advanced cancer stages, with reduced LDL bioavailability for chemical or oxidative modification [158]. In this context, SREBP-2-mediated LDLR negative feedback mechanism is lost in prostate cancer cells which accumulate large amounts of LDL [159]. Jiang et al. [160] propose that high oxLDL levels stimulate ROS levels which damages host cells, whilst low oxLDL levels stimulate VEGF-A synthesis, triggering tumor angiogenesis. This may explain why lower oxLDL concentrations are detected in advanced tumors.

LOX-1-induced upregulation of pro-angiogenic molecules, including VEGF-A, matrix metalloproteinases such as MMP-2 and MMP-9, can promote tumor metastasis [57]. Furthermore, oxLDL stimulates gene expression linked to cell cycle control [161] with implications for cancer progression [103]. Low levels of oxLDL (0.1–1 μg/mL) promotes mitosis and proliferation in ovarian cancer cells; it also decreases cisplatin treatment efficacy with an increased IC_50_ for biological activity [162]. Increased resistance to cancer chemotherapy linked to oxLDL levels is reported in osteosarcoma and multiple myeloma [163]. How oxLDL promotes cancer progression in this context remains to be fully understood.

### 5.2. Regulation of Epithelial–Mesenchymal Transition by oxLDL

Another signaling pathway linking atherogenesis and tumor metastasis involves oxLDL-induced epithelial–mesenchymal transition (EMT) of cancer cells: here, epithelial cancer cells exhibit loss of apical-basal polarity and cell–cell adhesion whilst gaining stem cell-like invasive properties [164]. These new features enables epithelial cancer cells to migrate into the circulation (e.g., blood) overcoming local nutrient deficiencies by setting up secondary tumors at other sites [164]. González-Chavarría et al. [58] found that oxLDL-mediated LOX-1 activation in prostate cancer cells induces EMT through upregulation of the mesenchymal markers, e.g., N-cadherin, vimentin, Snail and Slug, with a concomitant reduction in epithelial markers, e.g., E-cadherin and plakoglobin. Such changes in epithelial cell adhesion can modulate actin dynamics and promote epithelial cancer cell migration and invasion [58].

The endothelial-to-mesenchymal transition (EndMT) is a risk factor for both atherosclerosis and cancer: oxLDL inhibits Snail degradation through a LOX-1 signaling pathway to promote EndMT. In this EndMT state, endothelial cells acquire myofibroblast-like properties with increased MMP secretion and atherosclerotic plaque instability [165]. Activation of the PI3K-Akt signal transduction pathway normally inhibits GSK-3β activity [166]; however oxLDL-regulated GSK-3β inhibition appears to be independent of the PI3K-Akt pathway [167]. Conversely, oxLDL stimulates LOX-1 signaling and promotes Akt and GSK-3β phosphorylation in gastric cancer cells [168]. The mechanism of oxLDL-regulated stabilization of Snail could involve different pathways in primary endothelial vs. cancer cells. Inhibition of PI3K-Akt signaling partially suppresses oxLDL-induced EMT [168], suggesting additional regulatory events (Figure 4).

OxLDL inhibits GSK-3β-mediated phosphorylation, ubiquitination and proteasomal degradation of Snail, inducing EMT [168]. NF-κB activation increases COP9 signalosome 2 (CSN2) expression, which also inhibits Snail ubiquitination and degradation (Figure 4) [169]. CD36-oxLDL complex formation and signaling promotes focal adhesion kinase 1 (FAK1) activation and VAV1-mediated activation of Ras-related C3 botulinum toxin substrate and GTPase (Rac1), which inhibits non-muscle myosin II [170]. This subsequently promotes changes in cell morphology, namely loss of polarity and actin polymerization, which causes cell migration [171]. Rac1 also stimulates NADPH oxidase, increasing reactive oxygen species production and thus enhancing LDL oxidation and accelerating cancer progression. Fatty acid metabolites which activate PPARγ, including 9-hydroxyoctadecadienoic acid (HODE) and 13-HODE, are delivered to the cell following CD36 activation by oxLDL. This stimulates PPARγ which upregulates CD36, promoting EMT further [170]. FAK1 activation promotes increased actin dynamics and cell migration and contributes to cancer cell spread and invasion [172]. In macrophages, oxLDL-induced FAK1 activation alongside inactivation of Src homology domain 2 protein phosphatase (SHP2) inhibits cell migration; increased residence within pro-inflammatory microenvironments could further promote atherosclerosis (Figure 4) [173]. FAK1 activation thus has implications for both atherosclerosis and cancer.

The LOX-1-oxLDL pathway also activates protein kinase C (PKC) signal transduction, stimulating endothelial MMP secretion and EndMT; this may involve TGFβ [167,174]. Although oxLDL-induced EMT is complex depending on the cell type or tissue, Snail is a central regulatory factor [167]. Pro-atherogenic electronegative LDL and desialylated LDL also triggered EndMT whereas LDL did not [167]. However, circulating LDL undergoes multiple modifications in vivo, thus deciphering the biological role of each modified LDL species is currently not feasible [175]. OxLDL promotes microRNA-210 (*miR-210*) expression and cell migration (Figure 5) [176]. Elevated *miR-210* expression is not only detected in human atherosclerotic plaques [177] but also in colorectal [178], gastric [179] and lung [180] cancers and associated with poor breast cancer prognosis [181]. Elevated *miR-210* levels have been noted in mouse gastrointestinal tumors [182].

Methylation of the hypoxia-response element (HRE) in the *miR-210* promoter prevents HIF-1α binding and subsequent *miR-210* gene transcription. However, oxLDL decreases DNA methyltransferase 3b (DNMT3b) activity, causing hypomethylation of the *miR-210* promoter. This upregulates *miR-210* gene transcription by HIF-1. OxLDL also upregulates HIF-1α expression, thus further promoting *miR-210* gene transcription [182]. Expression of *miR-210* inhibits Sprouty-related EVH1 domain 2 (SPRED2) expression; this protein modulates cell migration via inhibition of oxLDL-mediated ERK activation and downstream c-Fos phosphorylation which impacts on MMP secretion. *MiR-210* down-regulates PDK1 expression which inhibits the PI3K-Akt-mTOR signaling pathway; this promotes endothelial apoptosis [168]. In macrophages, *miR-210* reduces expression of 2,4-dienoyl-CoA reductase (DECR1), a rate-limiting enzyme for β-oxidation of fatty acids and provision of metabolic energy. Such inhibition promotes macrophage necroptosis, contributing to the necrotic core of atherosclerotic plaques [183]. However, one report [184] suggests unstable atherosclerotic plaques display low *miR-210* expression. *MiR-210* targets the adenomatous polyposis coli (*APC*) mRNA 3′-UTR, reducing APC protein levels in VSMCs, stimulating the Wnt-β-catenin axis to promote VSMC survival. Therefore, oxLDL-induced *miR-210* upregulation may contribute to initial stages of atherosclerosis development but promotes plaque stability in late stages. The mechanism underlying *miR-210* overexpression has not been fully elucidated; further studies are required to understand how long non-coding and circular RNA expression, e.g., *miR-210* impact on CVD [185]. The potential of plaque-stabilizing agents, such as *miR-210* mimics or ultrasound-mediated nanoparticle technology, could reduce CVD mortality but requires care with the pro-tumorigenic effect of elevated *miR-210* levels [186].

## 6. OxLDL Modulates Immune Function, Autophagy and Cancer Survival

The oxLDL-induced neutrophil extracellular trap (NET) involving endothelial cells also promotes EMT. During NET, neutrophils release elastase which promotes proteolysis of cell surface endothelial VE-cadherin, an intercellular junction protein, causing β-catenin signaling and Snail (*SNAI1*) gene transcription [187]. Increased incidence of oxLDL and activated neutrophils are detected in high grade vs. low grade metastatic tumors [142]. One explanation is that neutrophil recruitment to the pro-inflammatory tumor microenvironment promotes LDL oxidation through myeloperoxidase release [188]. Biglycan expression on the surface of tumor endothelial cells (TECs) has high LDL affinity. Accumulated LDL within the tumor microenvironment is modified by ROS released by tumor cells [91]. ROS-induced LOX-1 upregulation on TECs facilitates oxLDL recognition and promotes monocyte chemoattractant protein 1 (MCP-1) production and release. MCP-1 binds to CCR2 on the surface of neutrophils and cancer cells and promotes migration towards endothelial cells. Neutrophil activation by oxLDL promotes neutrophil extracellular traps (NETs) with elevated myeloperoxidase levels, further promoting LDL oxidation and LOX-1 [89]. LOX-1 knockdown in tumor endothelial cells reduces MCP-1 gene expression, neutrophil recruitment and mitigated lung metastasis [89]. However, this study fails to explain how *LOX-1* gene transcription is regulated under pathophysiological conditions. As epigenetic dysregulation of another biomarker, biglycan, appears to promote tumor metastasis, future studies should investigate whether LOX-1 upregulation occurs via a similar mechanism [92].

High prevalence of neutrophils is associated with reduced overall survival in many solid tumors [189]; however, whether this association is reflective of neutrophil contribution to cancer progression, or is a secondary effect of inflammation in late stage cancer is unclear. Other studies report a protective role of neutrophils against cancer, including the induction of a T-cell response [190,191]. One explanation for these differing effects is the shift in neutrophil response from anti-cancer to pro-cancer during disease progression [192]. It was recently shown that oxidized and desialylated LDL suppresses anti-tumor responses by lymphokine-activated killer (LAK) cells, e.g., cell cytotoxicity and IFNγ production. Conversely, nLDL increased LAK cytotoxicity whilst oxidation and desialylation alone had no significant effects [193]. Modification of LDL promotes both pro-atherogenic events and cancer progression.

OxLDL-induced autophagy is linked to VSMC fate during atherosclerosis. Low oxLDL concentrations promote protective VSMC autophagy and cell proliferation, whereas autophagy-dependent cell death occurs at high concentrations, indicating that the autophagy-induced stress response malfunctions at a specific degree of cell injury [194]. Autophagy is also a key feature of cancer progression. Autophagy inhibition via knockdown of mediators of autophagy, Beclin1 and ATG5, suppresses hepatocellular carcinoma (HCC) metastasis in mouse models [195]. One component of oxLDL, 7-ketocholesterol, activates PPARγ which also causes elevation in proline oxidase (POX) levels. POX expression promotes ROS production which consequently stimulates Beclin-1 expression and autophagy [196]. However, autophagy inhibition did not modulate HCC cell migration or invasion, suggesting a role in cell survival under low nutrient and oxygen conditions [195]. Depending on cell type and disease stage, autophagy can also protect against cancer by limiting ROS production and DNA damage [197]. Further understanding the mechanism of oxLDL-regulated autophagy is needed in the context of atherosclerosis and cancer [197].

## 7. Anti-Atherogenic HDL Particles and Disease Role(s)

HDL levels inversely correlate with atherosclerosis risk and CVD [198]. HDL binding to Class B SR-B1 promotes reverse cholesterol transport and activation of the PI3K-Akt signaling pathway. SR-B1-HDL signaling promotes phosphorylation and activation of eNOS and increased nitric oxide (NO) synthesis, highlighting the anti-atherogenic properties of HDL [199]. HDL is a natural antioxidant by ROS neutralization, and transfer of oxidation-prone moieties from LDL to HDL for catabolism [200]. However, genetic studies fail to unambiguously confirm HDL as a protective factor against CVD [201,202]. Furthermore, therapeutic strategies aimed at increasing HDL levels, independent of LDL, have so far failed [203]. One explanation is that during metabolic disease states, structural modification of ApoA-1 associated with HDL causes the loss of vasoprotective and anti-atherosclerotic properties causing HDL to become a pro-inflammatory factor [204]. For example, ApoA-1 can undergo oxidation by myeloperoxidase within human atheromas [205], impairing HDL functionality, including cholesterol efflux capacity and antioxidant properties [206]. CAD patients exhibit modified HDL with LOX-1 binding properties; this stimulates endothelial PKCβII signaling and suppresses eNOS activation and NO production [207]. It is becoming clear that HDL functionality, rather than circulating levels, confers anti-atherogenic properties. Focus has therefore shifted away from promoting HDL levels to improving HDL anti-atherogenic capacity [208].

Administration of reconstituted ApoA-1 promotes cholesterol efflux in a Phase II clinical trial [209]. Bioengineered rice milk containing recombinant ApoA-1 administered to transgenic mice caused ~50% atherosclerotic plaque reduction [210]. Improving HDL antioxidant capacity has also been considered, but studies have primarily focused on in vitro oxidation [200]. The difficulties in assessing and comparing native and modified HDL antioxidant activities and functional outcomes means therapeutic strategies are hampered in this context [200]. Further work is required for the importance of HDL antioxidant capacity and whether promoting this aspect, e.g., dietary anthocyanin supplementation, is beneficial for vascular disease status [211].

Native HDL levels are inversely associated with cancer risk and can be viewed as an anti-cancer factor [212]. However, whether this is causal or consequential is unclear; SR-B1 is overexpressed in various cancer cell types which promotes HDL uptake to meet the lipid demands of cancer cells for growth, thereby depleting plasma HDL [213]. The increased levels of oxHDL in atherosclerosis further points to the close link between atherosclerosis and cancer. OxLDL and glycated HDL activate canonical MAPK, p38 MAPK and PI3K-Akt signal transduction pathways causing increased breast cancer cell migration and proliferation [214]. Furthermore, HDL levels are potentially reduced by high pro-inflammatory cytokine levels in cancer which prevent hepatic ApoA-1 expression [215].

HDL and synthetic HDL-like particles are potential delivery vehicles for hydrophobic, anti-cancer drugs with increased efficacy and specificity. The smaller HDL particle size of 5–15 nm diameter (vs. 25–30 nm diameter for LDL) facilitates HDL diffusion into capillaries and vessel walls [212]. Furthermore, SR-B1-mediated HDL recognition facilitates deliveries of small molecules and nucleic acids directly into the cytoplasm; however, LDL-containing cargo(es) are delivered to the lysosome [216].

### 7.1. Lipid-Lowering Dual Anti-Atherogenic and Anti-Cancer Therapy

Statins are the most widely used class of lipid-lowering drugs in the context of combatting atherosclerosis and arterial disease. Large randomized clinical trials show a significant reduction in the mortality of cardiovascular disease patients upon statin therapy [217]. Statins are inhibitors of cellular HMG-CoA reductase activity by blocking early steps in endogenous cholesterol biosynthesis [218]. Surprisingly, statins are also competitive inhibitors of oxLDL binding to the LOX-1 C-type lectin-like domain, i.e., CTLD [79]; furthermore, statins act to disrupt lipid rafts implicated in LOX-1-mediated oxLDL uptake and trafficking [219]. Inhibition of HMG-CoA reductase activity also reduces the levels of isoprenoids used for small G-protein (e.g., Ras, Rho) prenylation and membrane anchoring; aberrant activation of these signaling GTPases are linked to cell proliferation and migration in many cancer states [220]. Statin-mediated down-regulation of GTPase activity could suppress cancer initiation and progression [158]. Furthermore, statin inhibition of NF-κB activation and signaling reduces transcription of pro-angiogenic genes e.g., *VEGFA*, decreased tumor growth, increased tumor necrosis and apoptosis [221]. Furthermore, statins reduce breast, colorectal, ovarian, pancreatic and lung cancer risk [220] and inhibit tumor metastasis in animal models [158].

Statins were developed as cholesterol-lowering drugs to treat atherosclerosis and arterial disease; however, statin-based anti-cancer therapy requires much higher drug concentrations. Whereas cholesterol-lowering statin therapy requires sub-micromolar (<1 μM) concentrations, anti-tumor activity could require 10-fold higher statin levels (>10 μM) [222]. Another caveat is that statin activation of the pregnane X receptor (PXR) and constitutive androstane receptor (CAR) can promote multidrug resistance activity and chemoresistance [222]. Nonetheless, clinical trials have found that statin use as adjuvant therapy alongside chemotherapy for cancer is beneficial [223]. Cholesterol is also an agonist of PXR and CAR activity; statin-mediated LDL-C reduction likely counteracts statin-induced chemo-resistance [222]. Bioinformatics analysis of biomarkers of statin-responsive cancer therapy may allow the development of personalized cancer treatments where statin therapy is selectively used [224]. Cholesterol homeostasis is mediated by complex feedback mechanisms; thus inhibition of a single metabolic pathway for cholesterol may have limited impact on cancer progression [158]. As novel inhibitors of cholesterol metabolism are being developed, combination therapy to target cholesterol biosynthesis and esterification should be considered [158]. Future studies may also explore the potential for metabolic disease therapies to be repurposed for anti-cancer use [158].

Nearly 20% of the human population exhibit statin sensitivity including muscle lysis; in a search for new anti-hyperlipidemia therapies, PCSK9 inhibitors are being increasingly used [35]. The PCSK9 membrane protease binds to LDLR, promoting its lysosomal degradation and thereby increasing circulating bloodstream LDL levels, with increased arterial LDL deposition promoting atherosclerosis [35]. In MI patients, combining statin and humanized monoclonal antibody (mAb) anti-PCSK9 therapy increased plaque stabilization and regression compared to statin therapy alone [225]. As LDLR interacts with the CD8+ T-cell receptor complex and promotes TCR recycling and signaling, PCSK9 binding to LDLR reduces TCR recycling, signaling and cytotoxic T-lymphocyte anti-tumor activity [226]. Functional inhibition of CSK9 using either mAb or siRNA therapy used alongside immune checkpoint therapy could provide more effective anti-cancer therapy [227]. However, very low LDL levels may be associated with increased risk of cancer risk; longer-term studies are needed to determine the effects of LDL or cholesterol-lowering therapy in cancer care [228,229].

### 7.2. EMT Regulation and Disease Therapy

Activation of TGFβ and the canonical MAPK signal transduction pathways regulates EMT, VSMC proliferation and biglycan synthesis: targeting this pathway could suppress atherosclerosis and cancer progression [230]. Resveratrol, a natural polyphenolic compound, inhibits lung cancer metastasis by suppression of TGFβ-mediated EMT and promotes melanoma cell apoptosis by inhibiting the ERK-PKM2-Bcl-2 signaling axis [231,232]. In VSMCs, resveratrol inhibits gene expression linked to signaling pathways including FAK1 and Rac1; one consequence is reduced lamellipodia and cell migration [233]. Resveratrol therapy to inhibit signaling and cell migration could reduce both atherosclerosis and cancer progression. EMT is a key aspect of cancer metastasis and drug resistance and targeting this process could improve disease outcomes [234]. However, resveratrol usage is limited by rapid metabolism and clearance, with low solubility, absorption rate and bioavailability [230]. These issues could be overcome using modified resveratrol derivatives, bio-enhancers and nanodrug delivery [235,236]. Purification of plant-based resveratrol is limited by low final purity; expression in engineered microorganisms or chemical synthesis are alternative routes [237]. The mechanism(s) of resveratrol action have not been fully elucidated, with a lack of long-term clinical trials to address patient safety concerns [230].

### 7.3. Anti-Angiogenic Strategies in Disease Therapy

Anti-angiogenic therapy to treat tumor growth and metastasis is increasingly widespread. The use of humanized antibodies, synthetic proteins, modified nucleic acids and small molecule inhibitors have been approved for clinical use. However, such treatments are limited by their inability to provide long-term disease remission; furthermore, drug resistance and increased tumor growth are frequently detected. Anti-angiogenic drugs are frequently used as part of a multimodal therapy, where chemotherapy or other treatments are used simultaneously to provide the most effective outcomes. Blocking angiogenesis in the context of atherosclerosis could be either beneficial or potentially dangerous. Angiogenesis and the growth of blood vessel collaterals within the atherosclerotic plaque can contribute to plaque rupture and thrombosis. However, VEGF-regulated signaling pathway inhibitors used in anti-cancer therapy to target VEGFR tyrosine kinase activity can cause cardiovascular toxicity and promote atherosclerosis [238]. However, the increasing number of VEGF-targeted therapies means that there is scope for assessing drug efficacy in atherosclerosis and arterial disease states such as tumor angiogenesis.

### 7.4. Targeting Neutrophil Extracellular Traps (NETs)

NET formation requires chromatin decondensation requiring nuclear peptidylarginine deiminase 4 (PAD4) activity which mediates histone citrullination; PAD4 is thus a therapeutic target in inhibiting EMT and LDL oxidation [239]. The PAD4 inhibitors Cl-amidine and BB-Cl-amidine caused a reduction in disease burden in mouse models of atherosclerosis and cancer [240]; however, such therapy needs evaluation in human clinical trials [241,242]. Neutrophil elastase inhibitors are currently in clinical trials on bronchiectasis patients with promising results [243]; however, such drugs have yet to be tested in CVD and cancer patients. Currently no myeloperoxidase inhibitors with potency in human studies have been found [244]. Recently, natural guaiacol derivates have been identified which inhibit myeloperoxidase activity and LDL oxidation in vitro [245]. These compounds have good biocompatibility and low toxicity and are thus excellent candidates for clinical development and trials.

## 8. NADPH Oxidase and Anti-Oxidant Therapy

Targeting NADPH oxidase (NOX) activity for CVD therapy aims to reduce endothelial dysfunction caused by ROS and also suppress hydrogen peroxide-induced activation of neutrophil elastase which promotes nuclear decondensation in NET [242]. Apocynin is a plant-derived organic compound is which inhibits NOX activity and ROS production; it inhibits atherosclerosis, but reduced ROS levels could down-regulate immune surveillance and thus increase pathogen susceptibility [246]. The role of different NOX isoforms in cancer could lead to the use of NOX isoform-specific inhibitors, e.g., ML-171, a phenothiazine derivative which inhibits colon cancer invadopodia and cell migration [247]. Pharmacophore-based drug design could help to identify new NOX inhibitors that target several aspects of atherosclerosis and cancer [248].

Anti-oxidant (AOX) vitamins (e.g., vitamin E) and lipophilic AOX compounds (e.g., probucol) reduce ROS levels and LDL oxidation, inhibiting atherosclerosis in animal models [249]. However, many AOX therapies have been ineffective in human clinical trials [250,251,252]. One likelihood is that AOX therapy is a preventive or prophylactic treatment, such use after an acute clinical event may be unable to reverse disease progression or outcomes [249]. During atherosclerosis, plaque rupture and thrombosis, oxPL bound to human plasminogen decreases clot lysis times. Such fibrinolysis could counterbalance the pro-atherogenic oxLDL effects; this could be one reason AOX therapy is not conclusive in atherosclerosis [253]. AOX compounds which cross into the mitochondrial space and modulate oxidative events exhibit greater efficacy; this may be a better therapeutic avenue [249].

## 9. Metabolomics, Lipidomics and Biomarker Discovery

Both inflammatory and oxidative conditions are major risk factors for cancer development and progression. LOX-1 is functionally linked to pathophysiological processes such as atherogenesis, hypertension, tumorigenesis and metabolic dysfunction [46,57]. Metabolomics, the science of small biological molecule detection and analysis, offers a powerful approach for disease biomarker discovery, especially related to SRs such as LOX-1 and CD36. Metabolite profiling can identify potential biomarkers that reflect SR function linked to downstream effects; these can be valuable for diagnosis, prognosis and therapeutic monitoring [63,153,254,255]. Serum LOX-1 (sLOX-1) is a potential biomarker for acute coronary syndromes (ACS), T2D, stroke and metabolic syndromes such as obesity: there is potential for integrating such biomarker use into routine clinical practice and patient monitoring [256,257,258]. One challenge is the identification of biomarkers most suitable for effective, non-invasive and cost-effective use in clinical care and personalized medicine. By integrating biomarker profiling for both cancer and atherosclerosis, there may be flexibility in providing different monitoring technologies depending on disease state, progression, therapy and clinical care [256,258,259].

CAD is characterized by the presence of atherosclerotic plaques in the coronary arteries of the heart. The lack of cost-effective strategies for CAD monitoring reveals a pressing need for effective biomarkers which reflect disease severity and progression, both for reducing healthcare costs and improving patient outcomes. There are >1000 potential CAD biomarkers including genome-based, nucleic acid-derived, protein-based, carbohydrate-derived, and lipid-derived molecules or signatures (Table 2) [260,261]. Myoglobin, fatty acid binding proteins, and glycogen phosphorylase isoenzyme BB are early biomarkers, cardiac troponins T and I are late biomarkers (Table 2). Creatine kinase-myocardial band is an ACS biomarker that appears within 10–12 h of the acute clinical event [261].

Lipidomics is more recent but one of the most rapidly applied technologies in biomedical research. Here, different technical approaches are used to analyze lipid properties in relation to cell, tissue and animal physiology. Mass spectrometry (MS)-based lipidomics can monitor molecular changes in individual lipid species by identifying isomers, adducts and modifications [262]. Inflammation and disease status have major effects on LP status, including LP levels, lipid and lipoprotein chemical modifications. Detecting subtle changes in lipid and lipoprotein levels and structure means that conventional techniques are not applicable, requiring lipidomics-related techniques. In this context, development and application of a liquid chromatography-based MS/MS approach enables accurate quantitation of serum ApoA-1 and ApoB-100 biomarkers [263]. Such techniques could facilitate better CVD risk assessment in, e.g., dyslipidemia patients [264]. The identification and application of metabolic biomarkers in health and disease states requires new and advanced high throughput screening, MS-based methodologies and artificial intelligence (AI)-driven data analyses; integration of these different approaches will enable better healthcare delivery previously not thought to be feasible.

Metabolomics enables the identification of >1000 metabolites using high throughput screening techniques [265]. Generation of large metabolomic datasets are increasingly scrutinized using AI and machine learning (ML) algorithms. AI applications will enhance biomarker discovery since such programs can process large datasets to look for molecular signatures and statistically significant changes [266]. ML models allow disease classification by state, risk and prediction; ML can be used to model personalized responses and molecular signatures upon treatments. Such tools smoothen processing of data, enhance biomarker accuracy, allowing better personalized health assessment by integrating metabolomics datasets into other -omics techniques, e.g., genomics, proteomics [267].

## 10. Repurposing Existing Drugs and Therapeutic Limitations

There are some cardiometabolic drugs which share therapeutic efficacy across cancer and CVD. Statins, which are widely used to reduce the LDL-C, can help to lower TG levels and shows minimal effect on lipoprotein(a) levels [268]. Statins may also exhibit anti-cancer properties due to their cholesterol-lowering properties and ability to enhance immune checkpoint inhibitor (ICI) efficacy in cancer therapy [269,270]. Statin-mediated immunomodulatory and anti-proliferative effects in viral K-RAS transformed cells [271].

Inhibition of PCSK9 is used for hyperlipidemia patients where statin therapy is not suitable due to muscle lysis (~20% humans). PCSK9 targeting using siRNA (Inclisiran) or humanized monoclonal antibody (Evolocumab) to inhibit PCSK9 levels or activity, can boost LDL-R levels, clearance of circulating LDL, with lowered LDL-C [272,273,274]. PCSK9 inhibitors enhance the anti-cancer ICI therapy, e.g., PD-1 or PD-L1 blockage, via reduced LDL-C and TGF-β levels [275].

Metformin therapy is an established treatment for T2DM patients who have elevated serum glucose and insulin levels which could promote cell proliferation and cancer risk [276]. Using therapy to reduce baseline glucose levels in T2DM patients has wider benefits on cancer-linked energy metabolism, cellular growth, angiogenesis and apoptosis [277]. Metformin exerts both insulin-independent and insulin-dependent effects on cancer cells. However, recent clinical trials on metformin use in prostate cancer [278,279] and breast cancer [280] patients failed to provide significant benefit for disease progression and outcomes. This has led to the idea of metformin therapy as an immuno-metabolic adjuvant or prophylactic in cancer treatment. Preclinical studies shows metformin therapy alters the tumor immune microenvironment [281] and promotes PD-L1 degradation [282], both anti-cancer outcomes.

Glucagon-like peptide 1 receptor agonists (GLP-1 RAs) are incretin-based therapies which enhance insulin secretion after meals [283]. GLP-1 RAs promote weight loss, reduce chylomicron secretion and lower blood pressure [284]. GLP-1 RA treatment is associated with cancer risk reduction including colorectal, gallbladder, esophageal, liver, kidney, pancreatic and ovarian cancers; beneficial effects for meningioma and multiple myeloma are also noted. Using GLP-1 RA or metformin vs. insulin, can reduce CRC and gall bladder cancer risk [285].

Sodium-glucose co-transporter 2 (SGLT2) inhibitors such as empagliflozin, canagliflozin, dapagliflozin, and sotagliflozin are used to manage glucose levels in CVD and T2DM patients. The 2023 ESC guidelines denote that SGLT2 inhibitor use that can reduce acute CVD events, regardless of HbA1c levels and other glucose-lowering treatments, is recommended [286]. SGLT2 inhibitors show anti-proliferative effects against certain tumors, by inhibiting glucose uptake in metabolically reprogrammed cancer cells expressing SGLT2; preclinical studies show that the anti-cancer effects of SGLT2 inhibitor therapy are multifactorial, involving several metabolic pathways [287]. SGLT2 inhibitors helps to protect against cancer therapy-induced cardiovascular toxicity with cardioprotective effects against anthracycline exposure [288], and anti-cancer drug ponatinib-induced cardiotoxicity [289]. Patients with T2DM undergo cancer therapy with anthracyclines [290]; SGLT2 inhibitors can thus improve cancer patient care where cardiac dysfunction or heart failure is evident [291].

However, we lack solid evidence to support the efficacy of therapeutic intervention in subclinical atherosclerosis. To date, studies of subclinical atherosclerosis have typically focused on middle age or older populations (~40–70 yr age range). Subclinical atherosclerosis in younger individuals may be occurring, but we lack the tools to detect early signs of atherosclerosis and lesion development. Modifiable risk factors such as diet, alcohol intake and smoking can influence both therapeutic regimes and CVD outcomes. Beneficial diet regimes, e.g., Mediterranean diet, can help to lower CVD and cancer incidence, but individual, genetic and environmental variability alongside modifiable risk factors can interact and influence overall health and disease outcomes.

## 11. Conclusions and Future Perspectives

The failure of AOX therapy for CVD, stroke and other forms of arterial disease highlights our lack of mechanistic understanding of the molecular events underlying oxidative modifications, LP recognition and metabolism in health and disease states [249]. Emerging evidence points to a new role for SR-B1 in LDL transcytosis across the endothelium and deposition in blood vessel walls [292]; such trafficking could involve caveolin-1 and lipid rafts [293] alongside the protein kinase, ALK1 [294]. Aggregated LDL is a highly potent pro-atherogenic factor compared to native LDL or oxLDL; this promotes macrophage-linked fat and lipid deposits in early fatty streaks [295]. Although such findings highlight the causality of LDL in atherosclerosis, the proof for oxLDL as a driving force in atherosclerosis is still limited despite significant work using cell and animal models [252]. Much more studies are needed, e.g., on the adaptive immune response, and the roles of native LDL vs. oxLDL during atherosclerosis [296].

Chylomicrons were previously believed to have a limited role in atherosclerosis due to their larger size preventing entry to the intima [297]. However, as treatment with statins and PCSK9 inhibitors leaves a large residual CVD risk, the involvement of triglyceride-rich chylomicron, VLDL and their remnant particles in atherosclerosis is being reconsidered [298]. Genetic and epidemiological research has identified triglyceride-rich lipoproteins (TGRL) as causal factors for residual CVD risk [299,300]. In vitro studies have shown the involvement of VLDL and chylomicron particles in inflammation and oxidative stress [298]. For example, VLDL upregulates ICAM-1 and MCP-1 in ECs via NF-κB activation and chylomicron remnants activate monocytes and promote their migration [301,302]. Notably, chylomicron remnants and VLDL are directly internalized by macrophages without prior modification [298]. In fact, oxidized VLDL (oxVLDL) may be anti-atherogenic. Although VLDL oxidation induces a two-fold increase in its uptake by macrophages, oxVLDL lipolysis by lipoprotein lipase is reduced and therefore the triglyceride accumulation within macrophages is halved compared to native VLDL [303]; however, this work failed to establish the extent of VLDL oxidation in vivo. Conversely, it was found that VLDL glycation and oxidation reduces its binding to heparin and consequently prolongs its circulation in the blood [304]. As a result, triglycerides are transferred from VLDL to LDL, forming highly pro-atherogenic sdLDL particles. Furthermore, desialylation of VLDL promotes particle aggregation and cholesterol accumulation in SMCs [305]. Therefore, future studies should aim to clarify the net impact of VLDL modifications on particle atherogenicity. Heparin-based affinity chromatography can isolate heparin-bound (non-atherogenic) and non-bound (pro-atherogenic) subsets of VLDL. Therefore, whether it has potential use as a diagnostic tool warrants further study [304].

Using a breast cancer cell model, it was found that VLDL increases Akt phosphorylation and the subsequent induction of EMT, thereby promoting metastasis [104]. VLDL also suppressed anchorage-dependent cell death, whereas LDL and HDL did not. Therefore, the contribution of VLDL to cancer may be greater than that of LDL, however evidence is currently limited. Future studies investigating the potential of targeting TGRL levels to reduce CVD risk and metastasis, and greater understanding of the impact of VLDL modifications on tumorigenesis, are required [306]. The use of OMICs technologies may clarify the composition of TGRL particles and their remnants and clarify which particles are most significant to disease pathogenesis. This could enable the development of targeted therapies, such as the modulation of the angiopoietin-like family of glycoproteins which mediate lipoprotein lipase activity [298].

Currently, the LOX-1-oxLDL axis is an increasing area of focus in cancer; however understanding of its role in tumor stroma and the impact of oxLDL-induced EMT on cancer progression in vivo is limited [142]. Furthermore, although LOX-1 is overexpressed in various cancers and contributes to disease progression, future research is required to establish whether LOX-1 increases cancer risk and is a potential target for disease preventio [102]. As LOX-1 inhibitors are not presently available, the clinical effects of LOX-1 inhibition in humans is unclear [67]. Thus far, studies have largely focused on the role of HDL in cholesterol metabolism. However, HDL also impacts cellular serine production which is required for synthesis of glycine, proteins, lipids and nucleic acids, pathways important in cancer proliferation [212]. Therefore, further understanding of the involvement of HDL in other metabolic pathways is required.

In conclusion, the LOX-1-oxLDL axis explains, at least partially, the high degree of coexistence between atherosclerosis and cancer. Oxidative DNA damage, NET formation, EMT, angiogenesis and pro-inflammatory signaling via NF-κB activation are downstream effects of LOX-1 activation in both atherosclerosis and cancer. Recent studies have highlighted the outdatedness of the oxidative modification hypothesis; LDL and HDL are now understood to undergo various modifications prior to their oxidation which enhance their atherogenic and pro-tumorigenic properties. Therapeutics with the potential to target both diseases simultaneously through the LOX-1-oxLDL axis and improving HDL function have shown potential, however the requirement for high selectivity and the challenges in developing LOX-1 monoclonal antibodies have delayed advancements within the field.

## Figures and Tables

**Figure 1 biology-14-00675-f001:**
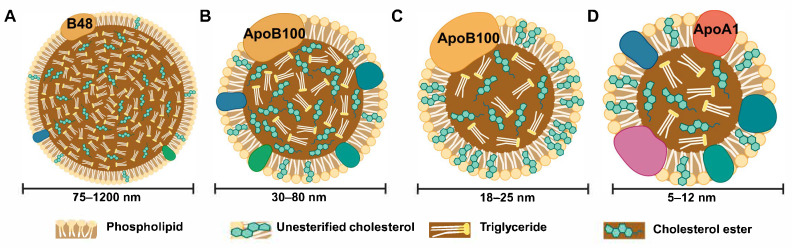
Lipoprotein particle structure. (**A**) Chylomicron, (**B**) very-low-density lipoprotein (VLDL), (**C**) low-density lipoprotein (LDL), and (**D**) high-density lipoprotein (HDL) particles. Generated using BioRender (www.biorender.com (accessed on 4 June 2025)).

**Figure 2 biology-14-00675-f002:**
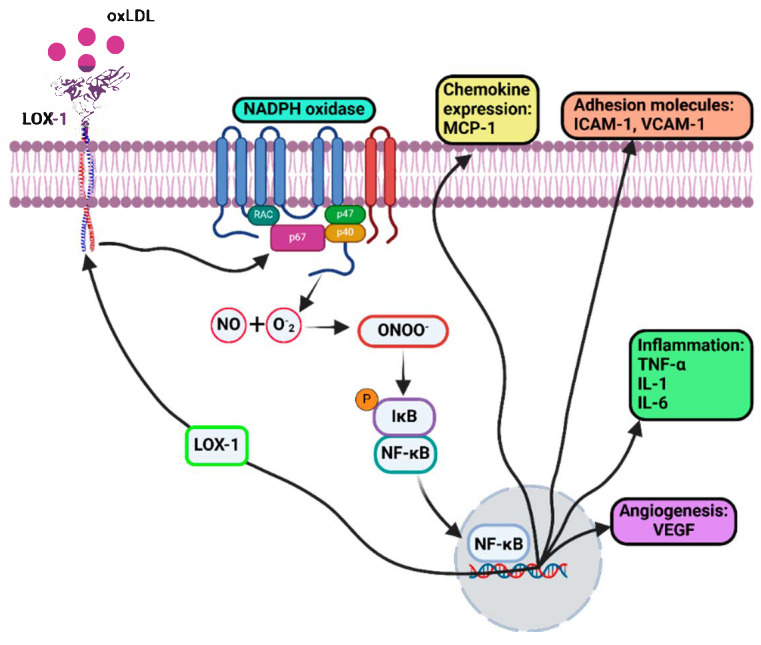
Regulation of NF-κB pro-inflammatory signaling by oxLDL. Schematic showing how LOX-1 and NADPH oxidase signaling regulates ROS production and pro-inflammatory signaling via NF-κB. The impact on other signaling pathways and gene expression are key features of such regulation. Generated using BioRender (www.biorender.com (accessed on 9 March 2025)).

**Figure 3 biology-14-00675-f003:**
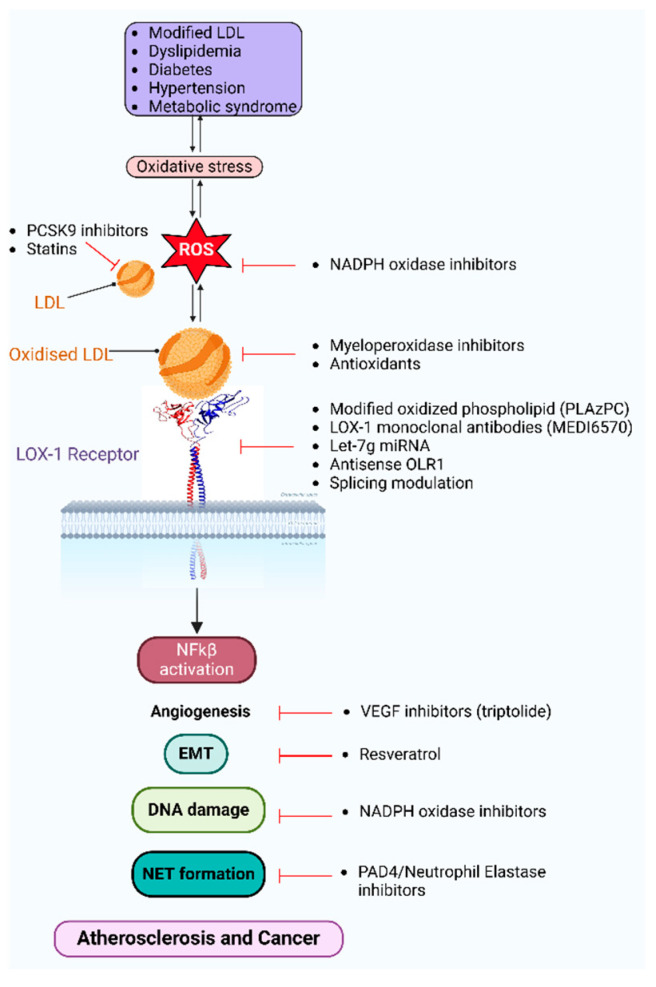
Therapeutic approaches targeting the LOX-1-oxLDL signaling pathway. Binding of a LOX-1 multimer to oxLDL causes receptor activation and signaling into the cell interior. Competitive inhibitors can block LOX-1 binding to oxLDL; however, down-regulation of LOX-1 expression using reverse genetics is a valid therapeutic approach. Finally, the use of membrane-permeable compounds to perturb LOX-1-oxLDL signaling in the cytosol could modulate molecular events associated with both CVD and cancer.

**Figure 4 biology-14-00675-f004:**
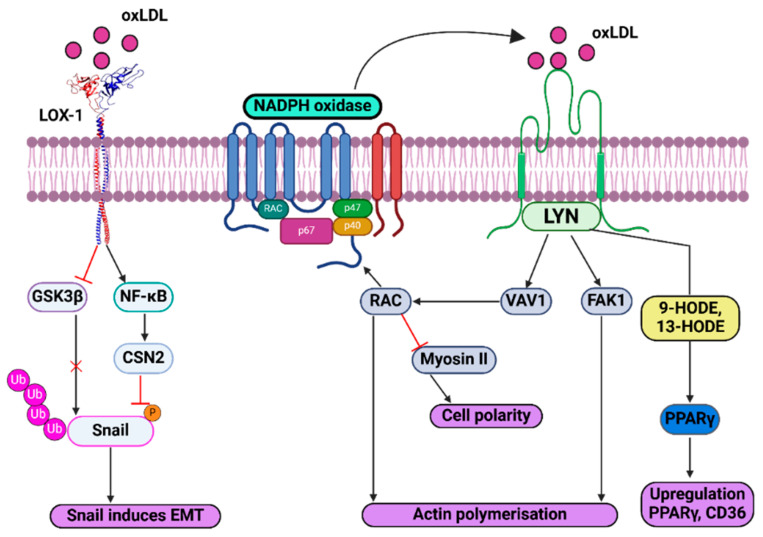
Epithelial–mesenchymal transition (EMT) regulation by oxLDL in cancer cells. Schematic depicting a functional link between LOX-1 activation and signaling on actin dynamics and cell migration. Such regulation impacts EMT and EndMT depending on the cell type, thus influencing cancer development and progression. Generated using BioRender (www.biorender.com (accessed on 9 March 2025)).

**Figure 5 biology-14-00675-f005:**
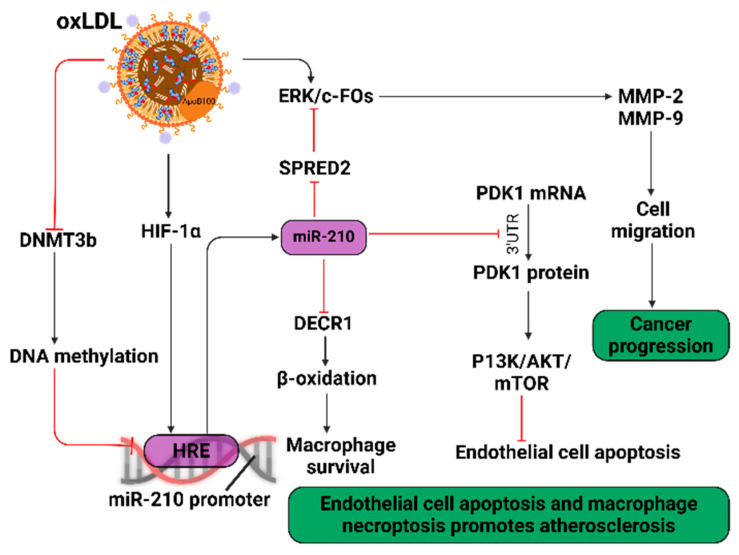
A role for *miR-210* in atherosclerosis and cancer. A schematic describing the link between microRNA-210, endothelial and immune cell function in contributing to atherosclerosis and cancer. Generated using BioRender (www.biorender.com (accessed on 9 March 2025)).

**Table 2 biology-14-00675-t002:** Established and emerging CVD biomarkers [260].

Protein-Based Biomarkers	
Aspartate Aminotransferase (AST)	Suppression of tumorigenicity 2 (ST2)
Lactate Dehydrogenase (LDH)	Matrix metalloproteinases (MMPs) Tissue inhibitors of metalloproteinases (TIMPs)
Carbonic Anhydrase III (CA-III)	Galectin-3
Brain Natriuretic Peptide (BNP and NT-pro BNP)	Procalcitonin
Mid-Regional Pro-a-type Natriuretic Peptide (MR-pro-ANP)	Myeloperoxidase (MPO)
Mid-Regional Pro Adrenomedullin (MR-proADM)	Fibrinogen
Endothelin-1 (CT-proCT-1)	Trimethylamine n-oxide (TMAO)
Creatine Kinases-MB (CK-MB)	Cystatin C
Hydroxy Butyrate Dehydrogenase (HBDH)	Myoglobin
Heart-fatty Acid Binding protein (H-FABP)	Ischemia-modified albumin (IMA)
Cardiac Troponins T, I (cTnT, cTnI)	Apoptosis antigen-1 (APO1/FAS)
C Reactive Protein (CRP)	Neutrophil gelatinase associated lipocalin (NGAL)
Tumor Necrosis Factor (TNF-α)	Uric acid (UA)
Interleukin-6 (IL-6)	Neuregulin-1 (NRG-1)
Pentraxin 3 (PTX-3)	Human serum albumin (HSA)
Pregnancy-associated plasma protein-A (PAPPA)	Serum amyloid A (SAA)
A soluble cluster of differentiation 40 ligand (sCD40L)	Retinol-binding protein 4 (RBP4)
Copeptin	Soluble lectin-like oxidized LDL receptor (sLOX-1)
Growth differentiation factor-15 (GDF-15)	Adiponectin (ADPN)
F2 Isoprostanes	S100 proteins

**Genome-based biomarker**	**Lipid biomarker**
Soluble ST2	Triglyceride:HDL-cholesterol ratio
Blood gene expression in CAD	LDL-cholesterol
DEFA1/DEFA3	Lipoprotein-associated phospholipase A2 (LP-PLA2)
	Oxylipin

**Pre-disease biological marker**	**Nucleic acid-derived Biomarker**
Hypertension (HTN)	microRNAs (miRNAs)

**Carbohydrate based biomarkers**	
Glycogen phosphorylase BB (GPBB)	

## Data Availability

Data is contained within the article.

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
