# Peer review of "Modified Lipid Particle Recognition: A Link Between Atherosclerosis and Cancer?"

_biology, 2025, doi:10.3390/biology14060675_

Round 1
Reviewer 1 Report
Comments and Suggestions for Authors
This review article explores the emerging connection between atherosclerosis and cancer through the lens of modified lipid particle recognition, particularly focusing on the LOX-1-oxLDL signaling axis. The authors effectively synthesize evidence from both cardiovascular and cancer research fields to demonstrate shared molecular mechanisms and signaling pathways that contribute to both diseases. The paper's main strength lies in its comprehensive integration of these typically separate research domains, offering valuable insights into potential therapeutic approaches that could simultaneously target both atherosclerosis and cancer.
General Concept Comments
The review provides a thorough examination of lipoprotein biology and the role of modified lipid particles in disease pathogenesis. However, several areas could be strengthened:
- The scope of the review could be expanded to include more recent developments in understanding lipid metabolism and its connection to disease, particularly given this is a 2025 publication.
- Cancer metabolism requires significant expansion: The manuscript inadequately addresses the critical connection between modified lipid particles and cancer cell metabolism. Recent studies have revealed how cancer cells reprogram lipid metabolism to support their growth and survival, with direct links to the LOX-1-oxLDL axis. The authors should include detailed discussion of how oxidized lipoproteins influence metabolic pathways in cancer cells, including altered fatty acid oxidation, enhanced lipogenesis, and metabolic adaptation. For example, M. Murdocca et al. (2021) "LOX-1, oxLDL, and PABPC1" provides important insights in this area that should be incorporated.
- Metabolomics and biomarker discovery section is missing: The manuscript lacks a dedicated section on recent advances in metabolomics and their application to identifying shared biomarkers between atherosclerosis and cancer. This represents a significant gap, as metabolite profiling has emerged as a powerful approach for early disease detection and risk stratification. The authors should incorporate studies showing how serum/plasma markers (including but not limited to LOX-1, oxLDL, and PABPC1) can serve as predictive biomarkers for both diseases. Key work such as Geng, S., et al. (2021) "Soluble LOX-1: A Novel Biomarker in Patients With Coronary Artery Disease, Stroke, and Acute Aortic Dissection?" should be discussed.
- Diet and lifestyle factors are overlooked: The manuscript fails to address how dietary patterns and lifestyle interventions affect lipid metabolism and modified lipoprotein formation in both disease contexts. There is no discussion of important clinical trials showing how dietary interventions (such as the Mediterranean diet or ketogenic diets) can simultaneously reduce oxLDL levels and cancer risk. This represents a significant missed opportunity to translate mechanistic insights into practical preventive strategies.
- Therapeutic implications need expansion: While the manuscript discusses some therapeutic approaches, it lacks comprehensive coverage of current and emerging therapies targeting LOX-1 and related metabolic pathways. The discussion should be expanded to include:
- Metabolic modulators affecting both diseases, such as those discussed in "Targeting AMPK for cancer prevention and treatment"
- Combination therapy approaches that could potentially address both pathologies simultaneously
Specific Comments
- Lines 135: While Figure 2 is visually informative, the figure could be improved by indicating potential points for therapeutic intervention in the pathway.
- Lines 182: The discussion of LOX-1's role in early stages of atherosclerosis mentions miR-let-7g but doesn't explore other microRNAs that may be involved.
- Lines 238: The section discussing oxLDL levels in early vs. advanced cancer stages presents a hypothesis about metabolic flux change in cancer cells. Including more metabolomics studies could provide stronger discussion.
- Lines 403: The paragraph starting at this line discusses the dual effects of HDL on both atherosclerosis and cancer, but the section title does not reflect this dual focus. The heading should be revised to better represent the content, perhaps to "Anti-atherogenic HDL particles and their role in cancer" rather than simply "Anti-atherogenic HDL particles."
- Lines 544: The section titled "LOX-1 RNA splicing and disease" disrupts the logical flow of the manuscript. This section appears abruptly following a comprehensive discussion of therapeutic approaches targeting the LOX-1-oxLDL signaling pathway, creating a disjointed transition. This content would be more effectively positioned earlier in the manuscript when LOX-1 biology and regulation are initially introduced, rather than inserted between therapeutic discussions and the subsequent CD36 targeting section.
- Lines 649: The conclusion section raises important points about remaining questions but doesn't provide a clear roadmap for future research priorities, particularly for cancer metabolism and therapeutic development targeting shared pathways.
- The manuscript needs a new section dedicated to cancer metabolic reprogramming in the context of modified lipid particles. This should include:
- How oxLDL influences the Warburg effect and mitochondrial metabolism in cancer cells
- The cross-talk between lipid metabolism and other metabolic pathways (glucose, amino acids)
- Metabolic vulnerabilities created by lipid metabolic alterations that could be therapeutically targeted
- The manuscript needs a new section dedicated to metabolomics approaches and biomarker discovery. This should include discussion of technologies like MS-based lipidomics, and machine learning applications for biomarker identification.
- A new section on diet and lifestyle interventions is needed. This should review landmark trials such as Estruch et al. (2022) on the Mediterranean diet's effects on cardiovascular outcomes and cancer risk, focusing on how dietary patterns modify lipoprotein profiles and oxidation states.
- The therapeutic section requires expansion to comprehensively cover:
- Metabolic modulators that target shared pathways
- Combination approaches targeting multiple aspects of lipid metabolism
- A table summarizing therapeutic approaches, their mechanisms, stage of development, and potential for dual disease benefit
In summary, while this is a well-structured review with several strengths, it requires significant expansion in the areas of cancer metabolism, metabolomics/biomarker discovery, diet/lifestyle interventions, and therapeutic approaches to achieve its full potential as a comprehensive review linking atherosclerosis and cancer through modified lipid particle recognition.
Author Response
Reviewer #1
This review article explores the emerging connection between atherosclerosis and cancer through the lens of modified lipid particle recognition, particularly focusing on the LOX-1-oxLDL signaling axis. The authors effectively synthesize evidence from both cardiovascular and cancer research fields to demonstrate shared molecular mechanisms and signaling pathways that contribute to both diseases. The paper's main strength lies in its comprehensive integration of these typically separate research domains, offering valuable insights into potential therapeutic approaches that could simultaneously target both atherosclerosis and cancer.
General Concept Comments
The review provides a thorough examination of lipoprotein biology and the role of modified lipid particles in disease pathogenesis. However, several areas could be strengthened:
- The scope of the review could be expanded to include more recent developments in understanding lipid metabolism and its connection to disease, particularly given this is a 2025 publication.
AU response: We have now corrected some references and modified the main manuscript by expanding the text to include more recent work on lipid metabolism, links to other diseases in the Introduction section (lines 93-134, pages 3-4) and a new Section 9 on metabolomics, lipidomics and biomarkers (lines 670-708).
- Cancer metabolism requires significant expansion: The manuscript inadequately addresses the critical connection between modified lipid particles and cancer cell metabolism. Recent studies have revealed how cancer cells reprogram lipid metabolism to support their growth and survival, with direct links to the LOX-1-oxLDL axis. The authors should include detailed discussion of how oxidized lipoproteins influence metabolic pathways in cancer cells, including altered fatty acid oxidation, enhanced lipogenesis, and metabolic adaptation. For example, M. Murdocca et al. (2021) "LOX-1, oxLDL, and PABPC1" provides important insights in this area that should be incorporated.
AU response: As per suggestion, we have substantially added to sections on cancer metabolism.
- 2. LOX-1, cancer and lipid metabolism (lines 210-226)
- Section 3. Warburg effect on cancer and CVD (lines 283-337)
- Metabolomics and biomarker discovery section is missing: The manuscript lacks a dedicated section on recent advances in metabolomics and their application to identifying shared biomarkers between atherosclerosis and cancer. This represents a significant gap, as metabolite profiling has emerged as a powerful approach for early disease detection and risk stratification. The authors should incorporate studies showing how serum/plasma markers (including but not limited to LOX-1, oxLDL, and PABPC1) can serve as predictive biomarkers for both diseases. Key work such as Geng, S., et al. (2021) "Soluble LOX-1: A Novel Biomarker in Patients With Coronary Artery Disease, Stroke, and Acute Aortic Dissection?" should be discussed.
AU response: We added a new Section 9 on metabolomics, lipidomics and biomarkers (lines 670-708).
- Diet and lifestyle factors are overlooked: The manuscript fails to address how dietary patterns and lifestyle interventions affect lipid metabolism and modified lipoprotein formation in both disease contexts. There is no discussion of important clinical trials showing how dietary interventions (such as the Mediterranean diet or ketogenic diets) can simultaneously reduce oxLDL levels and cancer risk. This represents a significant missed opportunity to translate mechanistic insights into practical preventive strategies.
AU response: We have now added information on lifestyle, diet and impact on lipid and metabolic changes linked to cancer and CVD. This is added in the text at:
- Introduction - Lifestyle effects including Mediterranean and ketogenic diets (lines 93-112)
- Therapeutic implications need expansion: While the manuscript discusses some therapeutic approaches, it lacks comprehensive coverage of current and emerging therapies targeting LOX-1 and related metabolic pathways. The discussion should be expanded to include:
- Metabolic modulators affecting both diseases, such as those discussed in "Targeting AMPK for cancer prevention and treatment"
- Combination therapy approaches that could potentially address both pathologies simultaneously
AU response: We have addressed the points raised by now adding new material in:
- Expanded section 2.3. LOX-1-specific therapeutics (lines 249-268) now including consideration of LOX-1-specific therapeutics in clinical trials
- New Table 1 on LOX-1 inhibitors (between lines 282-283)
- Section 9 on Metabolomics, lipidomics and biomarkers (lines 671-707)
- New Table 2 on biomarkers (between lines 707-710)
- Metabolic modulation in cancer and CVD is discussed in new section 3 (lines 283-337)
- Use of metabolic modulators in cancer and CVD is now addressed in new section 10 specifically on the use of drugs such as metformin (AMPK activator), GLP-1 RAs and SGLT-2 inhibitors (lines 710-742)
Specific Comments
- Lines 135: While Figure 2 is visually informative, the figure could be improved by indicating potential points for therapeutic intervention in the pathway.
AU response: This was previously Figure 3, now Figure 2 as we have moved it for better flow. In the new Figure 3, we clearly indicate that LOX-1 signaling to different events provides points of therapeutic intervention.
- Lines 182: The discussion of LOX-1's role in early stages of atherosclerosis mentions miR-let-7g but doesn't explore other microRNAs that may be involved.
AU response: We now clearly state:
- miR-let-7g in regulating LOX-1 expression (lines 198-201, 265-273; Fig. 3)
- Additional details on other potential miRNAs linked to LOX-1 expression and function (lines 269-273).
- miR-210 involvement in LOX-1 expression and function (lines 489-506)
- Lines 238: The section discussing oxLDL levels in early vs. advanced cancer stages presents a hypothesis about metabolic flux change in cancer cells. Including more metabolomics studies could provide stronger discussion.
AU response: We have substantially expanded our consideration of LOX-1 signaling and metabolic regulation in disease states. This is now linked to a broader discussion of metabolism and therapeutics through the manuscript.
- 2. LOX-1, cancer and lipid metabolism (lines 210—226, 280-282)
- New Table 1 on LOX-1 inhibitors (between lines 282-283)
- New material added to Section 5 on LOX-1 links to cancer (lines 416-427)
- Section 9 on Metabolomics, lipidomics and biomarkers (lines 671-707)
- New Table 2 on biomarkers (between lines 707-710)
- Metabolic modulation in cancer and CVD is discussed in new section 3 (lines 283-337)
- Use of metabolic modulators in cancer and CVD is now addressed in new section 10 specifically on the use of drugs such as metformin (AMPK activator), GLP-1 RAs and SGLT-2 inhibitors (lines 710-742)
- Lines 403: The paragraph starting at this line discusses the dual effects of HDL on both atherosclerosis and cancer, but the section title does not reflect this dual focus. The heading should be revised to better represent the content, perhaps to "Anti-atherogenic HDL particles and their role in cancer" rather than simply "Anti-atherogenic HDL particles."
AU response: We have now corrected this:
- Section 7. Anti-atherogenic HDL particles and disease role(s) (line 543)
- Lines 544: The section titled "LOX-1 RNA splicing and disease" disrupts the logical flow of the manuscript. This section appears abruptly following a comprehensive discussion of therapeutic approaches targeting the LOX-1-oxLDL signaling pathway, creating a disjointed transition. This content would be more effectively positioned earlier in the manuscript when LOX-1 biology and regulation are initially introduced, rather than inserted between therapeutic discussions and the subsequent CD36 targeting section.
AU response: We have now moved this as section 2.1 to lines 202-209.
- Lines 649: The conclusion section raises important points about remaining questions but doesn't provide a clear roadmap for future research priorities, particularly for cancer metabolism and therapeutic development targeting shared pathways.
AU response: We have improved the Section 11 on conclusions and future perspectives section by adding material by bringing together lipid metabolism, CVD, cancer and future therapeutics (lines 774-804). As previously mentioned more detailed therapeutic targeting of cancer and CVD is within the main manuscript text within sections 9 and 10.
- The manuscript needs a new section dedicated to cancer metabolic reprogramming in the context of modified lipid particles. This should include:
- How oxLDL influences the Warburg effect and mitochondrial metabolism in cancer cells
- The cross-talk between lipid metabolism and other metabolic pathways (glucose, amino acids)
- Metabolic vulnerabilities created by lipid metabolic alterations that could be therapeutically targeted
AU response: We now have expanded and added significantly in different parts of the manuscript in the following areas to address these points:
- 2. LOX-1, cancer and lipid metabolism (lines 210—226, 280-282)
- New Section 3. Warburg effect, cancer and CVD (lines 284-338)
- Section 10 - Glucose metabolism, metformin and other metabolic therapies (lines 721-743)
- Introduction – lipid metabolism linked to diet (lines 671-707)
- Metabolic modulation in cancer and CVD is discussed in new section 3 (lines 283-337)
- Use of metabolic modulators in cancer and CVD is now addressed in new section 10 specifically on the use of drugs such as metformin (AMPK activator), GLP-1 RAs and SGLT-2 inhibitors (lines 710-742)
- The manuscript needs a new section dedicated to metabolomics approaches and biomarker discovery. This should include discussion of technologies like MS-based lipidomics, and machine learning applications for biomarker identification.
AU response: This is a good point and we have now expanded the manuscript to include these issues:
- Section 9. Metabolomics, lipid metabolism and biomarker discovery (lines 672-708)
- Machine learning and new -omics approaches (lines 702-708)
- Table 2. CVD biomarkers (between lines 710-711).
- A new section on diet and lifestyle interventions is needed. This should review landmark trials such as Estruch et al. (2022) on the Mediterranean diet's effects on cardiovascular outcomes and cancer risk, focusing on how dietary patterns modify lipoprotein profiles and oxidation states.
AU response: Lifestyle, diet and metabolic changes linked to disease outcomes and therapies are now embedded within the revised manuscript at these points:
- Introduction - Lifestyle effects including Mediterranean and ketogenic diets (lines 93-112)
- Section 9. Metabolomics, lipid metabolism and biomarker discovery (lines 672-708)
- Section 10. Repurposing existing therapies (lines 711-752)
- LP profiles and oxidation states are now carefully discussed throughout the manuscript and the relationship to health and disease states
- The therapeutic section requires expansion to comprehensively cover:
- Metabolic modulators that target shared pathways
- Combination approaches targeting multiple aspects of lipid metabolism
- A table summarizing therapeutic approaches, their mechanisms, stage of development, and potential for dual disease benefit
AU response: We address these raised points in the following revisions:
- Section 1. Introduction – lipid metabolism, diet and disease linkage (lines 93-134)
- Section 2.2. LOX-1, cancer and lipid metabolism (lines 210—226, 280-282)
- Multiple disease therapy by targeting LOX-1 (lines 269-282)
- New Table 1 on LOX-1 inhibitors and effects on different disease-linked outcomes (between lines 282-283)
In summary, while this is a well-structured review with several strengths, it requires significant expansion in the areas of cancer metabolism, metabolomics/biomarker discovery, diet/lifestyle interventions, and therapeutic approaches to achieve its full potential as a comprehensive review linking atherosclerosis and cancer through modified lipid particle recognition.
Reviewer 2 Report
Comments and Suggestions for Authors
Biology 3602385
- The review paper titled "Modified lipid particle recognition: a link between atherosclerosis and cancer?" explores how LOX-1-mediated signaling pathways influence key cellular processes including proliferation, apoptosis, and inflammation within the context of both atherosclerosis and cancer. It also emphasizes the therapeutic potential of targeting LOX-1, as its elevated expression is associated with poor outcomes in both cardiovascular diseases and a range of cancers. Targeting LOX-1 could, therefore, offer a promising strategy for managing both disease conditions while they were intertwined.
- This review article is very important since the global burden posed by two interrelated, life-threatening diseases. By highlighting LOX-1 as a common driver of both atherosclerosis and cancer, the review paves the way for innovative therapeutic strategies that could address cardiovascular and oncologic outcomes simultaneously making the topic both clinically and biologically significant. This dual-targeted approach is especially valuable in aging populations, where the coexistence of heart disease and cancer is frequently observed. Identifying biomarkers like LOX-1 could enhance patient stratification and support more personalized treatment strategies, ultimately aiding in the prediction of therapy responses and the optimization of treatment plans.
- The manuscript is well-structured, and figures are neatly crafted and presented.
- The author should provide a more detailed discussion on the mechanisms by which elevated LOX-1 levels contribute to poor prognosis in prostate cancer or other cancer types. Additionally, it would be valuable to explore whether there are significant differences in these mechanisms when comparing cancers that arise in the context of atherosclerosis.
- The author should elaborate how epigenetic regulators regulate LOX-1 gene expression at both the DNA and RNA levels, along with the resulting biological outcomes. It would also be valuable to highlight any existing research that targets LOX-1 through the modulation of epigenetic modifiers, especially given the growing interest in epigenetic therapies (epi-drugs) within the fields of cardiovascular disease and cancer.
- The author should discuss whether any promising clinical studies, including ongoing or completed trials at any phase, have been reported that specifically target LOX-1 as a therapeutic approach for treating cancers associated with heart disease.
- The author should address how current technological limitations hinder the study of LOX-1 and modified lipids, particularly in the context of developing effective therapeutic strategies.
- The author should incorporate recent studies that explore the connection between LOX-1, cardiovascular disease, and cancer. It is hard to find up-to-date references on this topic in the manuscript.
- The author could further explore the potential of combination therapies targeting LOX-1, particularly through the repurposing of existing heart disease and cancer drugs, as a future direction in treatment strategies.
- The author should improve the clarity of the labeling, legends, and diagram in Figure 5, as they currently appear blurred.
- The author should discuss the limitations of the review.
- The author should consider adding a pictorial representation in the conclusion to visually illustrate the overall concept.
I recommend this article for acceptance after the author addresses all the above points.
Author Response
Reviewer #2
Biology 3602385
- The review paper titled "Modified lipid particle recognition: a link between atherosclerosis and cancer?" explores how LOX-1-mediated signaling pathways influence key cellular processes including proliferation, apoptosis, and inflammation within the context of both atherosclerosis and cancer. It also emphasizes the therapeutic potential of targeting LOX-1, as its elevated expression is associated with poor outcomes in both cardiovascular diseases and a range of cancers. Targeting LOX-1 could, therefore, offer a promising strategy for managing both disease conditions while they were intertwined.
- This review article is very important since the global burden posed by two interrelated, life-threatening diseases. By highlighting LOX-1 as a common driver of both atherosclerosis and cancer, the review paves the way for innovative therapeutic strategies that could address cardiovascular and oncologic outcomes simultaneously making the topic both clinically and biologically significant. This dual-targeted approach is especially valuable in aging populations, where the coexistence of heart disease and cancer is frequently observed. Identifying biomarkers like LOX-1 could enhance patient stratification and support more personalized treatment strategies, ultimately aiding in the prediction of therapy responses and the optimization of treatment plans.
AU response: In our revised manuscripts we have substantially added to the points raised in the following ways:
- Section 1. Introduction – lipid metabolism, diet and disease linkage (lines 93-134)
- Section 2.2. LOX-1, cancer and lipid metabolism (lines 210—226, 280-282)
- Multiple disease therapy by targeting LOX-1 (lines 269-282)
- New Table 1 on LOX-1 inhibitors and effects on different disease-linked outcomes (between lines 282-283)
- Section 9. Metabolomics, lipid metabolism and biomarker discovery (lines 672-708)
- New Table 2 on biomarkers (between lines 707-710)
- Section 10. Repurposing existing therapies (lines 711-752)
- The manuscript is well-structured, and figures are neatly crafted and presented.
AU response: Thank you for the kind comment.
- The author should provide a more detailed discussion on the mechanisms by which elevated LOX-1 levels contribute to poor prognosis in prostate cancer or other cancer types. Additionally, it would be valuable to explore whether there are significant differences in these mechanisms when comparing cancers that arise in the context of atherosclerosis.
AU response: We now discuss these issues in these sections:
- Section 2.2. LOX-1, cancer and lipid metabolism (lines 210—226, 280-282)
- Multiple disease therapy by targeting LOX-1 (lines 269-282)
- New Table 1 on LOX-1 inhibitors (between lines 282-283)
- New Section 3. Warburg effect, cancer and CVD (lines 284-338) links dysfunction in glucose and energy metabolism to cancer and CVD states, with links to lipid metabolism
- Section 10. Repurposing existing therapies (lines 711-752) considers metabolic therapy in cancer and CVD using drugs such as metformin (AMPK activator), GLP-1 RAs and SGLT-2 inhibitors (lines 710-742)
- The author should elaborate how epigenetic regulators regulate LOX-1 gene expression at both the DNA and RNA levels, along with the resulting biological outcomes. It would also be valuable to highlight any existing research that targets LOX-1 through the modulation of epigenetic modifiers, especially given the growing interest in epigenetic therapies (epi-drugs) within the fields of cardiovascular disease and cancer.
AU response: We have now added information about epigenetic modifications in this context (lines 413-427, 520-522).
- The author should discuss whether any promising clinical studies, including ongoing or completed trials at any phase, have been reported that specifically target LOX-1 as a therapeutic approach for treating cancers associated with heart disease.
AU response: We address this issue by adding these new materials:
- Section 2.2. LOX-1, cancer and lipid metabolism (lines 210—226, 280-282)
- 3. LOX-1-specific therapeutics (lines 232-282) including clinical trials on LOX-1-specific drug candidates especially for CVD
- Multiple disease therapy by targeting LOX-1 (lines 269-282)
- New Table 1 on LOX-1 inhibitors and effects on different disease-linked outcomes (between lines 282-283)
- LOX-1 links to cellular transformation (lines 397-406)
- LOX-1 links to cancer (lines 412-427, 437-442)
- LOX-1-oxLDL signaling links to EndMT, prostate and gastric cancer (lines 448-459)
- The author should address how current technological limitations hinder the study of LOX-1 and modified lipids, particularly in the context of developing effective therapeutic strategies.
AU response: We have now added information about the limitations on therapeutic approaches:
- Clinical trials on anti-LOX-1 and limitations (lines 249-268)
- Multiple disease therapy by targeting LOX-1 (lines 269-282)
- The link between LOX-1 expression, cancer and issues with dual disease anti-LOX-1 therapy (lines 437-442)
- LOX-1 links to cancer and limitations to our current understanding (lines 412-427,437-442, 448-459)
- Final 2 paragraphs in Section 11. Conclusions – how targeting the LOX-1-oxLDL axis could be beneficial but what our current limitations are (lines 789-804)
- The author should incorporate recent studies that explore the connection between LOX-1, cardiovascular disease, and cancer. It is hard to find up-to-date references on this topic in the manuscript.
AU response: We added new information on the links between CVD and cancer including LOX-1:
- Section 2. LOX-1 and atherosclerosis – expanded (lines 138-283)
- New Table 1 on LOX-1 inhibitors and effects on different disease-linked outcomes (between lines 282-283)
- Section 5. Links between cancer and atherosclerosis – expanded (lines 389-459)
- Section 9. Metabolomics, lipidomics and biomarker discovery – new section (lines 672-708)
- The author could further explore the potential of combination therapies targeting LOX-1, particularly through the repurposing of existing heart disease and cancer drugs, as a future direction in treatment strategies.
AU response: We have added new relevant material:
- New Section 10. Repurposing existing drug (lines 711-743)
- New 10.1. Therapeutic limitations (lines 745-752)
- The author should improve the clarity of the labeling, legends, and diagram in Figure 5, as they currently appear blurred.
AU response: We have redrawn all figures using BioRender and saved as .tiff files which are uploaded onto the resubmission separately in addition to the manuscript-embedded diagrams. If required, we can save these again at higher quality, but of course, each .tiff file size would increase.
- The author should discuss the limitations of the review.
AU response: We have added new relevant material:
- New 10.1. Therapeutic limitations (lines 745-752)
- Section 11. Conclusions and future perspectives (lines 754-804) – we have now expanded our long-term vision to include limitations and unknowns in our pathway to translation towards dual anti-cancer and anti-CVD therapy
- The author should consider adding a pictorial representation in the conclusion to visually illustrate the overall concept.
AU response: We have a new Graphical Abstract which is also uploaded separately as required by the Editorial Office:
- GA diagram in .tiff file
- GA diagram embedded in separate MS.docx file with Simple Summary
- GA diagram in main manuscript file
I recommend this article for acceptance after the author addresses all the above points.
Round 2
Reviewer 1 Report
Comments and Suggestions for Authors
Authors have substantially addressed all major concerns raised in the initial review. The significant improvements, especially the comprehensive cancer metabolism coverage that effectively bridges lipid metabolism with cancer progression and addition of metabolomics and biomarker discovery content providing practical research applications.
The manuscript now provides a valuable resource for researchers working at the cardiovascular-cancer interface, with strong mechanistic insights and translational relevance. It makes a significant contribution to understanding shared molecular pathways between these major diseases and will likely influence future research directions in this emerging field.